# The Number of Trials Matters in Infinite-Horizon General-Utility Markov Decision Processes

**Pedro P. Santos** [1,2] **Alberto Sardinha** [3] **Francisco S. Melo** [1,2]

## Abstract

The general-utility Markov decision processes (GUMDPs) framework generalizes the MDPs framework by considering objective functions that depend on the frequency of visitation of state-action pairs induced by a given policy. In this work, we contribute with the first analysis on the impact of the number of trials, i.e., the number of randomly sampled trajectories, in infinite-horizon GUMDPs. We show that, as opposed to standard MDPs, the number of trials plays a key-role in infinite-horizon GUMDPs and the expected performance of a given policy depends, in general, on the number of trials. We consider both discounted and average GUMDPs, where the objective function depends, respectively, on discounted and average frequencies of visitation of state-action pairs. First, we study policy evaluation under discounted GUMDPs, proving lower and upper bounds on the mismatch between the finite and infinite trials formulations for GUMDPs. Second, we address average GUMDPs, studying how different classes of GUMDPs impact the mismatch between the finite and infinite trials formulations. Third, we provide a set of empirical results to support our claims, highlighting how the number of trajectories and the structure of the underlying GUMDP influence policy evaluation.

## 1. Introduction

Markov decision processes (MDPs) (Puterman, 2014) provide a mathematical framework to study stochastic sequential decision-making. In MDPs, the agent aims to find a mapping from states to actions such that some function of the stream of rewards is maximized. The specification of the scalar reward function, which expresses the degree of desirability of each state-action pair, allows the encoding of different objectives. MDPs have found a wide range of applications in different domains (White, 1988), such as inventory management (Dvoretzky et al., 1952), optimal stopping (Chow et al., 1971) or queueing control (Stidham, 1978). MDPs are also of key importance in the field of reinforcement learning (RL) (Sutton & Barto, 2018) since the agent-environment interaction is typically formalized under the framework of MDPs. Recent years witnessed significant progress in solving challenging problems across various domains using RL (Mnih et al., 2015; Silver et al., 2017; Lillicrap et al., 2016). Such results attest to the flexibility of MDPs as a general framework to study sequential decision-making under uncertainty.

However, there are relevant objectives that cannot be easily specified within the framework of MDPs (Abel et al., 2022). These include, for example, imitation learning (Hussein et al., 2017; Osa et al., 2018), pure exploration problems (Hazan et al., 2019), risk-averse RL (García et al., 2015), diverse skills discovery (Eysenbach et al., 2018; Achiam et al., 2018) and constrained MDPs (Altman, 1999; Efroni et al., 2020). Such objectives, including the scalar reward objective of standard MDPs, can be formalized under the framework of general utility Markov decision processes (GUMDPs) (Zhang et al., 2020; Mutti et al., 2023). In GUMDPs, the objective is, instead, encoded as a function of the occupancy induced by a given policy, i.e., as a function of the frequency of visitation of states (or state-action pairs) induced when running the policy on the MDP. Recent works have unified such objectives under the same framework and proposed general algorithms to solve GUMDPs under convex objective functions (Zhang et al., 2020; Zahavy et al., 2021; Geist et al., 2022). Extensions to the case of unknown dynamics are also provided by the aforementioned works.

Despite providing a more flexible framework with respect to objective-specification in comparison to standard MDPs, Mutti et al. (2023) show that finite-horizon GUMDPs implicitly make an infinite trials assumption. In other words, GUMDPs implicitly assume the performance of a given policy is evaluated under an infinite number of episodes of interaction with the environment. Since this assumption may

[1]INESC-ID, Lisbon, Portugal [2]Instituto Superior Técnico, Lisbon, Portugal [3]PUC-Rio, Rio de Janeiro, Brazil. Correspondence to: Pedro P. Santos <pedro.pinto.santos@tecnico.ulisboa.pt>.

*Proceedings of the 42ⁿᵈ International Conference on Machine Learning*, Vancouver, Canada. PMLR 267, 2025. Copyright 2025 by the author(s).

be violated under many interesting application domains, the authors introduce a modification of GUMDPs where the objective function depends on the empirical state-action occupancy induced over a finite number of episodes. Under the introduced finite trials formulation, the authors show that the class of Markovian policies does not suffice, in general, to achieve optimality and that non-Markovian policies may need to be considered. Finally, the authors suggest that the difference between finite and infinite trials fades away under the infinite-horizon setting.

In this work, we contribute with the first analysis on the impact of the number of trials in infinite-horizon GUMDPs. We show that *the number of trials plays a key role in infinite-horizon GUMDPs*, as opposed to what has been suggested (Mutti et al., 2023). Such a finding is of interest and relevance as: (i) the infinite-horizon setting is one of the most prevalent settings in the planning/RL literature and has found important applications in different domains where the lifetime of the agent is uncertain or infinite; and (ii) the assumption that the agent is evaluated under an infinite number of trajectories is usually violated in relevant application domains. We focus our attention on discounted and average GUMDPs, where the objective function depends on discounted and average occupancies, respectively. We show, both theoretically and empirically, that the agent's performance may depend on the number of infinite-length trajectories drawn to evaluate its performance, but also on the structure of the underlying GUMDP. Our analysis fundamentally differs, from a technical point of view, from that in Mutti et al. (2023) where the authors consider the finite-horizon case; this is because discounted and average occupancies are inherently different than occupancies induced under the finite-horizon setting.

## 2. Background

Infinite-horizon MDPs (Puterman, 2014) provide a mathematical framework to study sequential decision making and are formally defined as a tuple $\mathcal{M} = (\mathcal{S}, \mathcal{A}, p, p_0, r)$ where: $\mathcal{S}$ is the discrete finite state space; $\mathcal{A}$ is the discrete finite action space; $p : \mathcal{S} \times \mathcal{A} \to \Delta(\mathcal{S})$ is the transition probability function with $\Delta(\mathcal{S})$ being the set of distributions over $\mathcal{S}$, $p_0 \in \Delta(\mathcal{S})$ is the initial state distribution; and $r : \mathcal{S} \times \mathcal{A} \to \mathbb{R}$ is the bounded reward function. The interaction protocol is: (i) an initial state $S_0$ is sampled from $p_0$; (ii) at each step $t$, the agent observes the state of the environment $S_t \in \mathcal{S}$ and chooses an action $A_t \in \mathcal{A}$. The environment evolves to state $S_{t+1} \in \mathcal{S}$ with probability $p(\cdot|S_t, A_t)$, and the agent receives a reward $R_t$ with expectation given by $r(S_t, A_t)$; (iii) the interaction repeats infinitely.

A decision rule $\pi_t$ specifies the procedure for action selection at each timestep $t$. A stochastic Markovian decision rule maps the current state to a distribution over actions, i.e., $\pi_t : \mathcal{S} \to \Delta(\mathcal{A})$. In the case of deterministic Markovian decision rules $\pi_t : \mathcal{S} \to \mathcal{A}$ instead. A policy $\pi = \{\pi_0, \pi_1, \ldots\}$ is a sequence of decision rules, one for each timestep. If, for all timesteps, the decision rules are deterministic or stochastic, we say the policy is deterministic or stochastic, respectively. We denote the class of Markovian policies with $\Pi_M$ and the class of Markovian deterministic policies with $\Pi_M^D$. A policy is stationary if it consists of the same decision rule for all timesteps. We denote with $\Pi_S$ the set of stationary policies and with $\Pi_S^D$ the set of stationary deterministic policies. We highlight that $\Pi_S \subset \Pi_M$ and $\Pi_S^D \subset \Pi_M^D$.

For a given policy $\pi$, the interaction between the agent and the environment gives rise to a random process $T = (S_0, A_0, S_1, A_1, \ldots) \in (\mathcal{S} \times \mathcal{A})^\mathbb{N}$, where the probability of trajectory $\tau = (s_0, a_0, s_1, a_1, \ldots)$ is given by $\zeta_\pi(\tau)$. In the case of $\pi \in \Pi_S$, we denote with $P^\pi$ the $|\mathcal{S}| \times |\mathcal{S}|$ matrix with elements $P^\pi(s, s') = \mathbb{E}_{A \sim \pi(\cdot|s)}[p(s'|s, A)]$.

**The infinite-horizon discounted setting**  The discounted state-action occupancy under policy $\pi$ is

$$d_{\gamma,\pi}(s, a) = (1 - \gamma) \sum_{t=0}^{\infty} \gamma^t \mathbb{P}_{\pi,p_0}(S_t = s, A_t = a), \quad (1)$$

where $\gamma \in [0, 1)$ is the discount factor and $\mathbb{P}_{\pi,p_0}(S_t = s, A_t = a)$ denotes the probability of state-action pair $(s, a)$ at timestep $t$ when following policy $\pi$ and $S_0 \sim p_0$. The expected discounted cumulative reward of policy $\pi$ can be written as $\langle d_{\gamma,\pi}, -r \rangle$[1] where $d_{\gamma,\pi} = [d_{\gamma,\pi}(s_0, a_0), \ldots, d_{\gamma,\pi}(s_{|\mathcal{S}|}, a_{|\mathcal{A}|})]$ and $r = [r(s_0, a_0), \ldots, r(s_{|\mathcal{S}|}, a_{|\mathcal{A}|})]$. We aim to find

$$\pi^* = \arg\min_{\pi \in \Pi_S} \langle d_{\gamma,\pi}, -r \rangle.$$

**The infinite-horizon average setting**  The average state-action occupancy under policy $\pi$ is

$$d_{\text{avg},\pi}(s, a) = \lim_{H \to \infty} \frac{1}{H} \sum_{t=0}^{H-1} \mathbb{P}_{\pi,p_0}(S_t = s, A_t = a). \quad (2)$$

The expected average reward of policy $\pi$ can be written as $\langle d_{\text{avg},\pi}, -r \rangle$, where $d_{\text{avg},\pi} = [d_{\text{avg},\pi}(s_0, a_0), \ldots, d_{\text{avg},\pi}(s_{|\mathcal{S}|}, a_{|\mathcal{A}|})]$. The analysis of the average reward setting depends on the structure of the Markov chains induced by conditioning the MDP on different policies. We say that an MDP is (Puterman, 2014; Altman, 1999):

- *unichain* if, for every $\pi \in \Pi_S^D$, the Markov chain with transition matrix $P^\pi$ contains a single recurrent class

---

[1] $\langle a, b \rangle$ denotes the dot product between vectors $a$ and $b$.

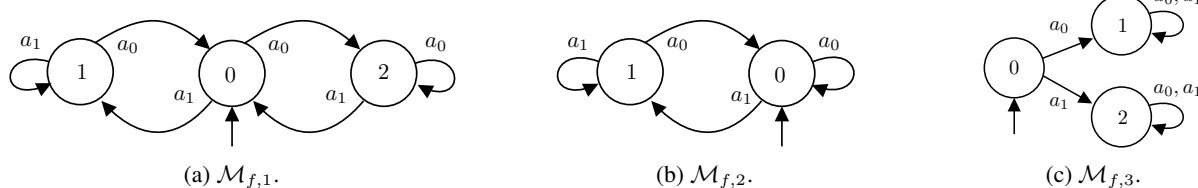

(a) $\mathcal{M}_{f,1}$.    (b) $\mathcal{M}_{f,2}$.    (c) $\mathcal{M}_{f,3}$.

Figure 1: Illustrative GUMDPs ($\mathcal{M}_{f,1}$ and $\mathcal{M}_{f,2}$ are adapted from Mutti et al., 2023). All GUMDPs have deterministic transitions. The objective for $\mathcal{M}_{f,1}$ is $f(d) = \langle d, \log(d) \rangle$ (entropy maximization), for $\mathcal{M}_{f,2}$ is $f(d) = \mathrm{KL}(d|d_\beta)$ where $\beta$ is a fixed policy (imitation learning), and for $\mathcal{M}_{f,3}$ is $f(d) = d^\top A d$ where $A$ is positive-definite (quadratic minimization).

plus a possibly empty set of transient states. A recurrent class is an irreducible class where all states are recurrent. An irreducible class is a set of states such that every state is reachable from any other state in the set. A state is recurrent if the probability of returning to it in the future is one and transient if the probability is less than one. In a finite-state Markov chain, all irreducible classes are recurrent.

- *multichain* if the MDP is not unichain.

Under both unichain and multichain MDPs we aim to find

$$\pi^* = \arg\min_{\pi \in \Pi_S} \langle d_{\mathrm{avg},\pi}, -r \rangle.$$

### 2.1. General-utility Markov decision processes

The framework of GUMDPs generalizes utility-specification by allowing the objective of the agent to be written in terms of the frequency of visitation of state-action pairs. This is in contrast to the standard MDPs framework, where the objective of the agent is encoded by the reward function.

We define an infinite-horizon GUMDP as a tuple $\mathcal{M}_f = (\mathcal{S}, \mathcal{A}, p, p_0, f)$ where $\mathcal{S}$, $\mathcal{A}$, $p$, and $p_0$ are defined in a similar way to the standard MDP formulation. The objective of the agent is encoded by $f : \Delta(\mathcal{S} \times \mathcal{A}) \to \mathbb{R}$, as a function of a state-action occupancy $d$. Similar to the case of standard MDPs, $d$ can correspond to: (i) a discounted state-action occupancy $d_\gamma$, as defined in (1), in the case of infinite-horizon discounted GUMDPs; or (ii) an average state-action occupancy $d_{\mathrm{avg}}$, as defined in (2), in the case of infinite-horizon average GUMDPs. The objective is then to find

$$\pi^* \in \arg\min_{\pi \in \Pi} f(d_\pi), \qquad (3)$$

where: (i) $d_\pi$ can either correspond to a discounted or average state-action occupancy depending on the considered setting; and (ii) $\Pi = \Pi_M$[2] in the case of average multichain GUMDPs and $\Pi = \Pi_S$ in the case of discounted GUMDPs

---

[2]Since $f$ can be non-linear it may happen that a stationary policy does not attain the minimum of $f$ and, hence, we need to consider the case where $\pi \in \Pi_M$ (Puterman, 2014, p. 402).

(Altman, 1999, Theo. 3.2.) and average unichain GUMDPs (Puterman, 2014, Theo. 8.9.4). When $f$ is linear, we are under the standard MDP setting; if $f$ is convex, we are under the convex MDP setting (Zahavy et al., 2021). Finally, it is known that the optimal policy may not be deterministic (Hazan et al., 2019; Zahavy et al., 2021).

**Illustrative GUMDPs**  Throughout this paper, we make use of the GUMDPs depicted in Fig. 1, which are representative of three common tasks in the convex RL literature.

## 3. From Expected to Empirical Objectives for GUMDPs

In this section, we introduce multiple objectives for GUMDPs. As opposed to the objective in (3), which depends on the *expected* discounted or average state-action occupancy $d_\pi$, the objectives herein introduced depend on the *empirical* discounted or average state-action occupancy.

We start by considering that the agent interacts with its environment over multiple trials, i.e., multiple trajectories/episodes. We denote by $K$ the number of trials. We assume the $K$ trials are independently sampled. As it is generally clear from the context to understand whether we are referring to discounted or average empirical occupancies, we use $\hat{d}$ to denote both types of empirical occupancies.

### 3.1. Empirical state-action occupancies

**Discounted state-action occupancies**  We introduce $\hat{d}_{\mathcal{T}_K}$, which denotes the empirical discounted state-action occupancy induced by a set of $K$ trajectories $\mathcal{T}_K = \{T_1, \ldots, T_K\}$, where each $T_k = (S_{k,0}, A_{k,0}, S_{k,1}, A_{k,1}, \ldots)$, defined as

$$\hat{d}_{\mathcal{T}_K}(s, a) = \frac{1}{K} \sum_{k=1}^{K} (1-\gamma) \sum_{t=0}^{\infty} \gamma^t \mathbf{1}(S_{k,t} = s, A_{k,t} = a),$$
(4)

where $\mathbf{1}$ is the indicator function. In practice, it is common to truncate the trajectories of interaction between the agent and its environment. We denote by $H \in \mathbb{N}$ the length at which the trajectories are truncated, i.e. the length of the

sampled trajectories. We then introduce a truncated version of estimator $\hat{d}_{\mathcal{T}_K}$, which we denote $\hat{d}_{\mathcal{T}_K,H}$, defined as

$$\hat{d}_{\mathcal{T}_K,H}(s,a) = \frac{1}{K} \sum_{k=1}^{K} \frac{1-\gamma}{1-\gamma^H} \sum_{t=0}^{H-1} \gamma^t \mathbf{1}(S_{k,t}=s, A_{k,t}=a). \tag{5}$$

**Average state-action occupancies** In the case of average state-action occupancies, we define

$$\hat{d}_{\mathcal{T}_K}(s,a) = \frac{1}{K} \sum_{k=1}^{K} \lim_{H\to\infty} \frac{1}{H} \sum_{t=0}^{H-1} \mathbf{1}(S_{k,t}=s, A_{k,t}=a). \tag{6}$$

We emphasize that the estimator above $\hat{d}_{\mathcal{T}_K}$ always considers infinite-length trajectories.

### 3.2. Infinite and finite trials objectives for GUMDPs

We now introduce multiple objectives for GUMDPs that are functions of empirical discounted/average state-action occupancies. Below, in the case of a discounted GUMDP, $\hat{d}_{\mathcal{T}_K}$ corresponds to an empirical discounted occupancy and $d_\pi = d_{\gamma,\pi}$. For average GUMPDs, $\hat{d}_{\mathcal{T}_K}$ is an empirical average occupancy and $d_\pi = d_{\text{avg},\pi}$.

The *finite trials* discounted/average objective, $f_K$, is

$$\min_\pi f_K(\pi) = \min_\pi \mathbb{E}_{\mathcal{T}_K}\left[f(\hat{d}_{\mathcal{T}_K})\right],$$

where $T_k \sim \zeta_\pi$ for each $T_k \in \mathcal{T}_K$. The *infinite trials* discounted/average objective, $f_\infty$, is

$$\min_\pi f_\infty(\pi) = \min_\pi f(d_\pi) = \min_\pi f\left(\mathbb{E}_{\mathcal{T}_K}\left[\hat{d}_{\mathcal{T}_K}\right]\right),$$

where $T_k \sim \zeta_\pi$ for each $T_k \in \mathcal{T}_K$. We note that $f_\infty$, under both discounted and average occupancies, is equivalent to the objective introduced in (3). Precisely, we call the objective above the infinite trials objective because, assuming $f$ is continuous, $\lim_{K\to\infty} f(\hat{d}_{\mathcal{T}_K}) = f(d_\pi) = f_\infty(\pi)$. The *finite trials truncated* objective, $f_{K,H}$, which we only consider under discounted occupancies, is

$$\min_\pi f_{K,H}(\pi) = \min_\pi \mathbb{E}_{\mathcal{T}_K}\left[f(\hat{d}_{\mathcal{T}_K,H})\right],$$

where $T_k \sim \zeta_\pi$ for each $T_k \in \mathcal{T}_K$. We note that the finite trials truncated objective is more general than the finite trials objective. In particular, $f_{K,H} = f_K$ as $H \to \infty$.

**Why there may be a mismatch between the infinite and finite trials objectives?** When $f$ is linear, we make the following remark.

*Remark* 3.1. If $f$ is linear, for both discounted and average occupancies, we have that $f_\infty(\pi) = f_K(\pi)$, for any $K \in \mathbb{N}$.

*Proof.* Under both discounted and average occupancies $\hat{d}$, for any $K \in \mathbb{N}$, it holds that

$$f_\infty(\pi) = \langle d_\pi, b\rangle = \langle \mathbb{E}_{\mathcal{T}_K}\left[\hat{d}_{\mathcal{T}_K}\right], b\rangle$$
$$= \mathbb{E}_{\mathcal{T}_K}\left[\langle \hat{d}_{\mathcal{T}_K}, b\rangle\right] = f_K(\pi),$$

due to the linearity of the expectation. $\square$

Thus, all objectives are equivalent. Intuitively, the performance of a given policy is, in expectation, the same independently of the number of trajectories drawn to evaluate its performance.

However, assume that the objective function $f$ is convex, possibly non-linear. We make the following remark.

*Remark* 3.2. If $f$ is convex, for both discounted and average occupancies, we have that $f_\infty(\pi) \leq f_K(\pi)$, for any $K \in \mathbb{N}$.

*Proof.* Under both discounted and average occupancies, for any $K \in \mathbb{N}$ and convex $f$, it holds that

$$f_\infty(\pi) = f(d_\pi) = f\left(\mathbb{E}_{\mathcal{T}_K \sim \zeta_\pi}\left[\hat{d}_{\mathcal{T}_K}\right]\right)$$
$$\leq \mathbb{E}_{\mathcal{T}_K \sim \zeta_\pi}\left[f\left(\hat{d}_{\mathcal{T}_K}\right)\right] = f_K(\pi),$$

where the inequality follows from Jensen's inequality. $\square$

As a consequence, the theorem above suggests that, in general, there may be a mismatch between the finite and infinite trials formulations for GUMDPs. In the next section we show that, indeed, $f_K(\pi) \neq f_\infty(\pi)$ in general and further investigate the impact of the number of trajectories in the mismatch between the infinite and finite trials formulations under both discounted and average occupancies.

## 4. Policy Evaluation in the Finite Trials Regime

In this section, we investigate the mismatch between the different GUMDP objectives introduced in the previous section, while evaluating the performance of a fixed policy. We consider convex objective functions. First, we focus our attention on the discounted setting and show that, in general, $f_K(\pi) \neq f_\infty(\pi)$, for fixed $\pi \in \Pi_S$. Furthermore, we provide a lower bound on the mismatch between $f_K(\pi)$ and $f_\infty(\pi)$, as well as an upper probability bound on the absolute distance between $f(\hat{d}_{\mathcal{T}_K,H})$ and $f_\infty(\pi)$. Second, we study policy evaluation under GUMDPs with average occupancies. We investigate the mismatch between $f_K(\pi)$ and $f_\infty(\pi)$ for different classes of GUMDPs, also proving a lower bound on the mismatch between $f_K(\pi)$ and $f_\infty(\pi)$. Finally, we provide a set of empirical results to support our theoretical claims.

## 4.1. The infinite-horizon discounted setting

We first consider the discounted setting. Thus, let $d_\pi = d_{\gamma,\pi}$, as defined in (1). Also, we consider estimators $\hat{d}_{\mathcal{T}_K}$ and $\hat{d}_{\mathcal{T}_K,H}$ as defined in (4) and (5) respectively. We prove the following result.

**Theorem 4.1.** *Under the discounted setting, it does not always hold that $f_K(\pi) = f_\infty(\pi)$ for arbitrary $\pi \in \Pi_S$.*

*Proof.* We prove the theorem by providing a GUMDP instance where $f_K(\pi) \neq f_\infty(\pi)$. We consider the GUMDP $\mathcal{M}_{f,3}$ (Fig. 1c). For simplicity, we let $f$ and the occupancies depend only on the states. Hence, $d = [d(s_0), d(s_1), d(s_2)]$. We let $f(d) = d^\top A d$, where $A$ is the identity matrix (hence, $f$ is a strictly convex function). It holds that $d_\pi = [(1-\gamma), \gamma\pi(a_0|s_0), \gamma\pi(a_1|s_0)]$. On the other hand, let $K = 1$. It holds that, with probability $\pi(a_0|s_0)$, the trajectory gets absorbed into $s_1$ and $\tau = (s_0, s_1, s_1, \ldots)$, yielding $\hat{d}_\tau = [(1-\gamma), \gamma, 0]$. With probability $\pi(a_1|s_0)$ the trajectory gets absorbed into $s_2$ and $\tau = (s_0, s_2, s_2, \ldots)$, yielding $\hat{d}_\tau = [(1-\gamma), 0, \gamma]$. Let $p = \pi(a_0|s_0)$ and note that $\pi(a_1|s_0) = 1 - p$. For any non-deterministic policy, i.e., $p \in (0, 1)$, it holds that

$$
\begin{aligned}
f_\infty(\pi) &= f(d_\pi) \\
&= f(p[(1-\gamma), \gamma, 0] + (1-p)[(1-\gamma), 0, \gamma]) \\
&< pf([(1-\gamma), \gamma, 0]) + (1-p)f([(1-\gamma), 0, \gamma]) \\
&= f_{K=1}(\pi),
\end{aligned}
$$

where the inequality holds since $f$ is strictly convex. $\qquad\square$

As stated in the theorem above, under the discounted setting, $f_K(\pi) \neq f_\infty(\pi)$ in general. Thus, we further analyze the impact of the number of trials, $K$, on the deviation between $f_K(\pi)$ and $f_\infty(\pi)$. To derive the result below, we assume $f$ is $c$-strongly convex, i.e., it exists $c > 0$ such that $f(d_1) \geq f(d_2) + \nabla f(d_2)^\top (d_1 - d_2) + \frac{c}{2}\|d_1 - d_2\|_2^2$, for any $d_1, d_2$ belonging to the domain of $f$. We note that the objective functions of all GUMDPs in Fig. 1 are $c$-strongly convex (proof in appendix). We state the following result (full proof in appendix).

**Theorem 4.2.** *Let $\mathcal{M}_f$ be a discounted GUMDP with $c$-strongly convex $f$ and $K \in \mathbb{N}$ be the number of sampled trajectories. Then, for any policy $\pi \in \Pi_S$ it holds that*

$$
\begin{aligned}
f_K(\pi) - f_\infty(\pi) &\geq \frac{c}{2K} \sum_{\substack{s \in \mathcal{S} \\ a \in \mathcal{A}}} \operatorname*{Var}_{T \sim \zeta_\pi} \left[\hat{d}_T(s, a)\right] \\
&= \frac{c(1-\gamma)^2}{2K} \sum_{\substack{s \in \mathcal{S} \\ a \in \mathcal{A}}} \operatorname*{Var}_{T \sim \zeta_\pi} \left[J_{r_{s,a}}^{\gamma,\pi}\right],
\end{aligned}
$$

*where $J_{r_{s,a}}^{\gamma,\pi} = \sum_{t=0}^\infty \gamma^t r_{s,a}(S_t, A_t)$ is the discounted return for the MDP with reward function $r_{s,a}(s', a') = 1$ if $s' = s$ and $a' = a$, and zero otherwise.*

*Proof sketch.* From the strongly convex assumption it holds, for a random vector $X$, that

$$
\mathbb{E}[f(X)] \geq f(\mathbb{E}[X]) + \frac{c}{2}\mathbb{E}\left[\|X - \mathbb{E}[X]\|_2^2\right].
$$

Using the inequality above, it holds that

$$
\begin{aligned}
f_K(\pi) - f_\infty(\pi) &= \mathbb{E}_{\mathcal{T}_K}\left[f(\hat{d}_{\mathcal{T}_K})\right] - f\left(\mathbb{E}_{\mathcal{T}_K}\left[\hat{d}_{\mathcal{T}_K}\right]\right) \\
&\geq \frac{c}{2}\mathbb{E}_{\mathcal{T}_K}\left[\left\|\hat{d}_{\mathcal{T}_K} - d_\pi\right\|_2^2\right] \\
&\overset{(a)}{=} \frac{c}{2}\sum_{s \in \mathcal{S}, a \in \mathcal{A}} \operatorname*{Var}_{\mathcal{T}_K}\left[\frac{1}{K}\sum_{k=1}^K \hat{d}_{T_k}(s, a)\right] \\
&= \frac{c}{2K}\sum_{s \in \mathcal{S}, a \in \mathcal{A}} \operatorname*{Var}_{T \sim \zeta_\pi}\left[\hat{d}_T(s, a)\right],
\end{aligned}
$$

where (a) follows from simplifying the previous expression and substituting $\hat{d}_{\mathcal{T}_K}(s, a) = \frac{1}{K}\sum_{k=1}^K \hat{d}_{T_k}(s, a)$ where $\hat{d}_{T_k}(s, a) = (1-\gamma)\sum_{t=0}^\infty \gamma^t \mathbf{1}(S_{k,t} = s, A_{k,t} = a)$. The result follows since $\hat{d}_T(s, a)$ is equivalent to the discounted return in an MDP with an indicator reward function. $\qquad\square$

As stated in the theorem above, the difference between $f_K(\pi)$ and $f_\infty(\pi)$ can be lower bounded by the sum of the variances of the discounted returns for the MDPs with reward functions $r_{s,a}$, as defined above. We refer to Benito (1982); Sobel (1982); Sitař (2006) for an expression to calculate the variance of discounted returns in MDPs. We highlight the $1/K$ dependence on the number of trajectories. Therefore, the result above shows that, for a low number of trajectories, the mismatch between the objectives can be significant, linearly decaying as $K$ increases.

Finally, we provide a probability bound on the absolute deviation between $f(\hat{d}_{\mathcal{T}_K})$ and $f_\infty(\pi)$, for fixed $\pi \in \Pi_S$. To derive our result, we assume $f$ is $L$-Lipschitz, i.e., $|f(d_1) - f(d_2)| \leq L\|d_1 - d_2\|_1$, for any $d_1, d_2$ belonging to the domain of $f$. We prove the following result (full proof in Appendix).

**Theorem 4.3.** *Let $\mathcal{M}_f$ be a discounted GUMDP with convex and $L$-Lipschitz $f$, $K \in \mathbb{N}$ be the number of sampled trajectories, each with length $H \in \mathbb{N}$. Then, for any policy $\pi$ and $\delta \in (0, 1]$ it holds with probability at least $1 - \delta$*

$$
|f_\infty(\pi) - f(\hat{d}_{\mathcal{T}_K,H})| \leq L\left(\sqrt{\frac{2|\mathcal{S}||\mathcal{A}|\log(2H/\delta)}{K}} + 2\gamma^H\right).
$$

*Proof sketch.* Via the application of successive inequalities, it can be shown that, for any $\pi$,

$$
|f_\infty(\pi) - f(\hat{d}_{\mathcal{T}_K,H})| \leq
$$
$$
L\left(\max_{t \in \{0,\ldots,H-1\}}\left\|\hat{d}_{K,t} - d_{\pi,t}\right\|_1 + 2\gamma^H\right). \quad (7)
$$

Using a union bound and the fact that $\mathbb{P}\left(\left\|d_{\pi,t} - \hat{d}_{K,t}\right\| > \epsilon'\right) \leq 2\exp\left(-\frac{1}{2|\mathcal{S}||\mathcal{A}|}K(\epsilon')^2\right)$ (Efroni et al., 2020, Lemma 16) we have that, with probability at least $1 - \delta$,

$$\max_{t \in \{0,\ldots,H-1\}} \left\|\hat{d}_{K,t} - d_{\pi,t}\right\|_1 \leq \sqrt{\frac{2|\mathcal{S}||\mathcal{A}|\log(2H/\delta)}{K}}.$$

Substituting the result above in (7) yields our result. $\qquad\square$

As shown above, for fixed $H \in \mathbb{N}$, the bound becomes arbitrarily tight up to a factor of $2L\gamma^H$ as we increase the number of sampled trajectories $K$; factor $2L\gamma^H$ is due to the bias of our estimator, which exponentially vanishes as $H$ increases. However, the bound highlights a $1/\sqrt{K}$ dependence on $K$, suggesting that for low $K$ values the mismatch between $f_\infty(\pi)$ and $f(\hat{d}_{\mathcal{T}_K,H})$ can become significant. Finally, the upper bound does not get tighter as $H$ increases, for fixed $K \in \mathbb{N}$.

In summary, our results under the discounted setting show that, indeed, a mismatch between $f_K$ and $f_\infty$ exists, as showcased in Theo. 4.1, where we provided a GUMDP under which $f_K \neq f_\infty$, and in Theo. 4.2, where we proved a lower bound on the deviation between $f_K$ and $f_\infty$. Finally, Theo. 4.3 further analyses how $f(\hat{d}_{\mathcal{T}_K,H})$ concentrates around $f_\infty(\pi)$ depending on the number of trajectories $K$, as well as the length of each trajectory $H$.

### 4.2. The infinite-horizon average setting

We now study the mismatch between the infinite and finite trials formulations of GUMDPs under the case of average occupancies. Hence, we consider estimator $\hat{d}_{\mathcal{T}_K}$ as defined in (6) and $d_\pi = d_{\text{avg},\pi}$. We always consider infinite-length trajectories. We investigate which properties of the GUMDP contribute to the mismatch between infinite and finite trials.

We start by focusing our attention on unichain GUMDPs, i.e., GUMDPs such that, for all $\pi \in \Pi_S^D$, the Markov chain with transition matrix $P^\pi$ has at most one recurrent class plus a possibly empty set of transient states. To prove the next result (full proof in appendix), we assume $f$ is continuous and bounded in its domain. All objective functions in Fig. 1 are continuous and bounded, however, for the case of the KL-divergence in $\mathcal{M}_{f,2}$ we need to ensure $d_\beta$ is lower-bounded to meet our assumptions.

**Theorem 4.4.** *If the average GUMDP $\mathcal{M}_f$ is unichain and $f$ is bounded and continuous in its domain, then $f_K(\pi) = f_\infty(\pi)$ for any $\pi \in \Pi_S$.*

*Proof sketch.* Consider the case where the occupancies are only state-dependent. For any $\pi \in \Pi_S$, in a unichain GUMDP, the Markov chain with transition matrix $P^\pi$ and initial distribution $p_0$ has a unique stationary distribution

$\mu_\pi \in \Delta(\mathcal{S})$. Let $Z_{H,k}$ be the random vector with components $Z_{H,k}(s) = \frac{1}{H}\sum_{t=0}^{H-1} \mathbf{1}(S_{k,t} = s)$. It holds that,

$$f_K(\pi) = \mathbb{E}_{\mathcal{T}_K}\left[f\left(\frac{1}{K}\sum_{k=1}^{K}\lim_{H\to\infty} Z_{H,k}\right)\right]$$
$$\stackrel{\text{(a)}}{=} f(\mu_\pi) = f_\infty(\pi),$$

where (a) holds because: (i) from the Ergodic theorem for Markov chains (Levin et al., 2006), $Z_{H,k} \to \mu_\pi$ almost surely $\forall k \in \{1, \ldots, K\}$; (ii) since $f$ is continuous, it also holds that $f(\frac{1}{K}\sum_{k=1}^{K} Z_{H,k}) \to f(\mu_\pi)$ almost surely; and (iii) from (ii) and the fact that $f$ is bounded, the bounded convergence theorem (Durrett, 2019, Theo. 1.6.7.) allows to simplify the expectation. We then generalize the result for the case of state-action dependent occupancies by considering a Markov chain defined over $\mathcal{S} \times \mathcal{A}$. $\qquad\square$

The result above states that, under unichain GUMDPs with continuous and bounded $f$, all objectives are equivalent. We now address the case of multichain GUMDPs, i.e., GUMDPs that are not unichain and, therefore, the Markov chain $P^\pi$ contains two or more recurrent classes.

**Theorem 4.5.** *If the average GUMDP $\mathcal{M}_f$ is multichain, then it does not always hold that $f_K(\pi) = f_\infty(\pi)$ for arbitrary $\pi \in \Pi_M$.*

*Proof.* We prove the theorem above by providing a GUMDP instance where $f_K(\pi) \neq f_\infty(\pi)$. We consider the GUMDP $\mathcal{M}_{f,3}$ (Fig. 1c), which is multichain. For simplicity, we let $f$ and the occupancies depend only on the states. Thus, $d = [d(s_0), d(s_1), d(s_2)]$. We let $f(d) = d^\top A d$, where $A$ is the identity matrix (hence, $f$ is a strictly convex function). It holds that $d_\pi = [0, \pi(a_0|s_0), \pi(a_1|s_0)]$. On the other hand, let $K = 1$. With probability $\pi(a_0|s_0)$, the trajectory gets absorbed into $s_1$ and $\tau = (s_0, s_1, s_1, \ldots)$, yielding $\hat{d}_\tau = [0, 1, 0]$. With probability $\pi(a_1|s_0)$ the trajectory gets absorbed into $s_2$ and $\tau = (s_0, s_2, s_2, \ldots)$, yielding $\hat{d}_\tau = [0, 0, 1]$. Let $p = \pi(a_0|s_0)$ and note that $\pi(a_1|s_0) = 1 - p$. For any non-deterministic policy, i.e., $p \in (0,1)$, it holds that

$$f_\infty(\pi) = f(d_\pi) = f([0, p, (1-p)])$$
$$= f(p[0, 1, 0] + (1-p)[0, 0, 1])$$
$$< pf([0, 1, 0]) + (1-p)f([0, 0, 1]) = f_{K=1}(\pi),$$

where the inequality holds since $f$ is strictly convex. $\qquad\square$

The result above shows that, under multichain GUMDPs, $f_K(\pi) \neq f_\infty(\pi)$ in general. The intuition is that each trajectory eventually gets absorbed into one of the recurrent classes and, therefore, multiple trajectories may be required so that $\hat{d}_{\mathcal{T}_K} \approx d_\pi$ and, hence, $f(\hat{d}_{\mathcal{T}_K}) \approx f(d_\pi)$. Therefore,

we now further investigate the mismatch between the finite and infinite objectives by proving a lower bound on the deviation between $f_K(\pi)$ and $f_\infty(\pi)$ while assuming $f$ is $c$-strongly convex (full proof in appendix).

**Theorem 4.6.** *Let $\mathcal{M}_f$ be an average GUMDP with $c$-strongly convex $f$ and $K \in \mathbb{N}$ be the number of sampled trajectories. Consider also the Markov chain with state-space $\mathcal{S}$, transition matrix $P^\pi$ and initial states distribution $p_0$. Let $\mathcal{R}$ be set of all recurrent states of the Markov chain and $\mathcal{R}_1, \ldots, \mathcal{R}_L$ the sets of recurrent classes, each associated with stationary distribution $\mu_l$. Then, for any policy $\pi \in \Pi_S$, it holds that*

$$f_K(\pi) - f_\infty(\pi) \geq$$
$$\frac{c}{2K} \sum_{l=1}^{L} Var_{B \sim Ber(\alpha_l)}[B] \sum_{s \in \mathcal{R}_l} \sum_{a \in \mathcal{A}} \pi(a|s)^2 \mu_l(s)^2,$$

*where $B \sim Ber(p)$ denotes that $B$ is distributed according to a Bernoulli distribution such that $\mathbb{P}(B = 1) = p$ and*

$$\alpha_l = \lim_{t \to \infty} \mathbb{P}(S_t \in \mathcal{R}_l | S_0 \sim p_0)$$

*is the probability of absorption to $\mathcal{R}_l$ when $S_0 \sim p_0$.*

*Proof sketch.* Let $\hat{d}_T(s,a) = \lim_{H \to \infty} Z_H(s,a)$ where $Z_H(s,a) = \frac{1}{H} \sum_{t=0}^{H-1} \mathbf{1}(S_t = s, A_t = a)$. Under a multichain GUMDP, $Z_H(s,a) \to Y_s \pi(a|s)$ almost surely where $Y_s$ is a random variable such that: (i) if $s$ is transient, $\mathbb{P}(Y_s = 0) = 1$; and (ii) if $s$ is recurrent, $Y_s = \mu_{l(s)}(s)$ with probability $\alpha_{l(s)}$ and $Y_s = 0$ with probability $1 - \alpha_{l(s)}$, where $l(s)$ denotes the index of the recurrent class to which state $s$ belongs. Then,

$$f_K(\pi) - f_\infty(\pi) = \mathbb{E}_{\mathcal{T}_K}\left[f(\hat{d}_{\mathcal{T}_K})\right] - f\left(\mathbb{E}_{\mathcal{T}_K}\left[\hat{d}_{\mathcal{T}_K}\right]\right)$$
$$\stackrel{(a)}{\geq} \frac{c}{2} \mathbb{E}_{\mathcal{T}_K}\left[\left\|\hat{d}_{\mathcal{T}_K} - d_\pi\right\|_2^2\right]$$
$$= \frac{c}{2K} \sum_{s \in \mathcal{S}, a \in \mathcal{A}} Var_{T \sim \zeta_\pi}\left[\hat{d}_T(s,a)\right]$$
$$\stackrel{(b)}{=} \frac{c}{2K} \sum_{s \in \mathcal{S}, a \in \mathcal{A}} \pi(a|s)^2 Var[Y_s]$$
$$\stackrel{(c)}{=} \frac{c}{2K} \sum_{l=1}^{L} Var_{B \sim Ber(\alpha_l)}[B] \sum_{\substack{s \in \mathcal{R}_l \\ a \in \mathcal{A}}} \pi(a|s)^2 \mu_l(s)^2$$

where: (a) follows from the $c$-strongly convex assumption; in (b) we used the fact that $Z_H(s,a) \to Y_s \pi(a|s)$ almost surely and the bounded convergence theorem (Durrett, 2019, Theo. 1.6.7.) to simplify the variance term; and (c) follows from rewriting $Y_s$ using a Bernoulli random variable, noting that $Var(Y_s) = 0$ for transient states, and simplifying the resulting expression. $\square$

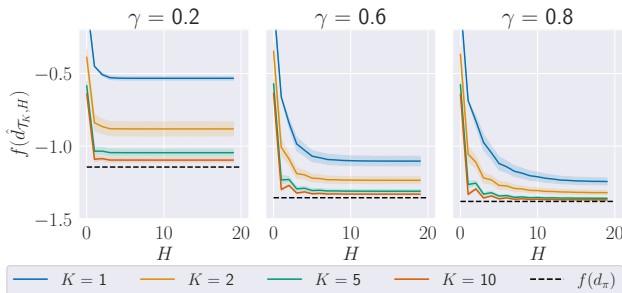

Figure 2: Empirical study of $f(\hat{d}_{\mathcal{T}_K,H})$ for different $K$, $H$ and $\gamma$ values under $\mathcal{M}_{f,1}$.

Intuitively, the result above shows that the gap between $f_K(\pi)$ and $f_\infty(\pi)$ can be lower bounded by a weighted sum of the variances of the probabilities of getting absorbed into each of the recurrent classes. Thus, we expect the gap to exist whenever the sampled trajectories can get absorbed into different recurrent classes. Also, we highlight the $1/K$ dependence on the number of sampled trajectories. Finally, we note that, in the case of unichain GUMDPs, since the unique recurrent class is reached with probability one, the lower bound above equals zero, agreeing with Theo. 4.4.

### 4.3. Empirical results

We now empirically assess the impact of different parameters in the mismatch between $f_{K,H}(\pi)$ and $f_\infty(\pi)$ for arbitrary fixed $\pi$. Under the discounted setting, we consider $\hat{d}_{\mathcal{T}_K,H}$ as defined in (5). We also use $\hat{d}_{\mathcal{T}_K,H}$ to study the average setting by letting $H \to \infty$ and $\gamma \to 1$, since $\lim_{H \to \infty} \lim_{\gamma \to 1} (5) = (6)$. We consider the GUMDPs depicted in Fig. 1. Under $\mathcal{M}_{f,1}$, $\pi(\text{left}|s_0) = \pi(\text{right}|s_0) = 0.5$, and $\pi(\text{right}|s_1) = \pi(\text{left}|s_2) = 1$; for $\mathcal{M}_{f,2}$ and $\mathcal{M}_{f,3}$, $\pi$ is uniformly random. We consider 100 random seeds and report 95% bootstrapped confidence intervals (shaded areas in plots). Complete experimental results are in the appendix. The code used can be found in the following repository.

**Discounted setting** ($\gamma < 1$) In Fig. 2, a set of plots displays the average finite trials objective function, $f(\hat{d}_{\mathcal{T}_K,H})$, in comparison to the infinite trials objective $f(d_\pi)$, under GUMDP $\mathcal{M}_{f,1}$. As can be seen, the results highlight that $f(\hat{d}_{\mathcal{T}_K,H})$ can significantly differ from $f(d_\pi)$. Also, for the displayed $\gamma$ values, both the trajectories' length $H$ and the number of trajectories $K$ need to be sufficiently high for the mismatch between $f(\hat{d}_{\mathcal{T}_K,H})$ and $f(d_\pi)$ to fade away. This is suggested by Theo. 4.3 since both $K$ and $H$ contribute to the tightness of the upper bound. We display the results for estimator $\hat{d}_{\mathcal{T}_K,H}$ under all GUMDPs in the appendix.

In Fig. 3a, we display a set of plots comparing $f(\hat{d}_{\mathcal{T}_K,H=\infty})$ and $f(d_\pi)$ for the different GUMDPs, under infinite-length trajectories. As can be seen when $\gamma < 1$, even for $H = $

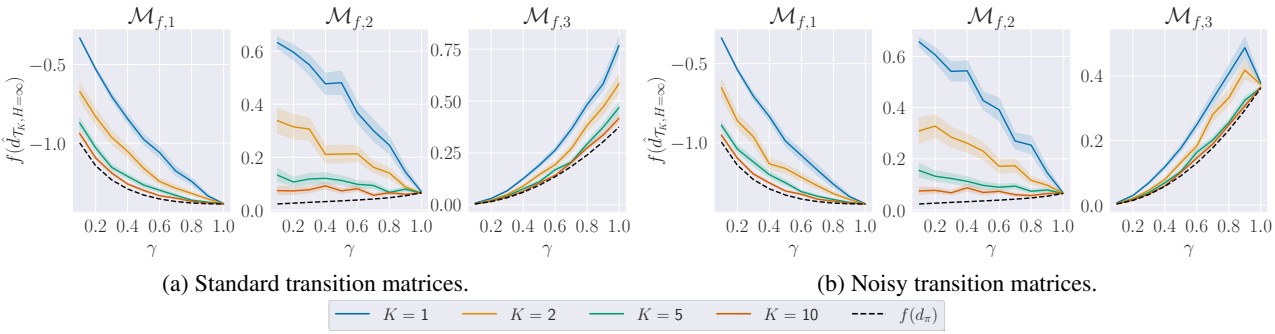

(a) Standard transition matrices.        (b) Noisy transition matrices.

Figure 3: Empirical study of $f(\hat{d}_{\mathcal{T}_K, H=\infty})$ for different $K$ and $\gamma$ values, with $H = \infty$.

$\infty$, $f(\hat{d}_{\mathcal{T}_K, H=\infty})$ can significantly differ from $f(d_\pi)$ if the number of sampled trajectories is low. Such results support the fact that $K$ plays a key role in regulating the mismatch between the different objectives for discounted GUMDPs and are in line with our theoretical analysis.

**Average setting ($\gamma \to 1$, $H \to \infty$)** As can be seen in Fig. 3a, under $\mathcal{M}_{f,1}$ and $\mathcal{M}_{f,2}$, we have that the difference between $f(\hat{d}_{\mathcal{T}_K, H=\infty})$ and $f(d_\pi)$ fades away when $\gamma \to 1$. However, this is not the case for $\mathcal{M}_{f,3}$. Under $\mathcal{M}_{f,1}$ and $\mathcal{M}_{f,2}$, we obtain such results because, given the choice of policies, the induced Markov chains have a single recurrent class. Hence, since there exists a unique stationary distribution, $\hat{d}_{\mathcal{T}_K, H=\infty}$ converges to $d_\pi$ irrespective of $K$ and the different objectives become equivalent. However, under $\mathcal{M}_{f,3}$ and for the chosen policy, the induced Markov chain has two recurrent classes and, hence, there exist multiple stationary distributions. Thus, a low number of trajectories ($K$ value) does not suffice to evaluate the non-linear objective.

Our results are in line with the theoretical results from the previous section. We note that all GUMDPs in Fig. 1 are multichain. Hence, from Theo. 4.5, it not always holds that $f(\hat{d}_{\mathcal{T}_K, H=\infty}) = f(d_\pi)$ in general. Our results for $\mathcal{M}_{f,3}$ exemplify that such a mismatch can occur. Naturally, being multichain does not imply that $f(\hat{d}_{\mathcal{T}_K, H=\infty}) \neq f(d_\pi)$ for: (i) all policies; or (ii) any policy. For (i), take our results under $\mathcal{M}_{f,1}$ as an example. For (ii), consider $\mathcal{M}_{f,2}$. For all policies except $\pi(\text{left}|s_1) = 1$ (zero otherwise) and $\pi(\text{right}|s_2) = 1$ (zero otherwise), the induced Markov chain has a single recurrent class and, hence, $f(\hat{d}_{\mathcal{T}_K, H=\infty}) = f(d_\pi)$. However, under the policy just described, even though the induced Markov chain has two recurrent classes, one of them is unreachable given the distribution of initial states and, hence, it also holds that $f(\hat{d}_{\mathcal{T}_K, H=\infty}) = f(d_\pi)$. Thus, for $\mathcal{M}_{f,2}$, the different objectives are equivalent for all policies.

**Average setting ($\gamma \to 1$, $H \to \infty$) with noisy transitions** We consider the GUMDPs in Fig. 1, but add a small amount

of noise to the transition matrices so that there is a non-zero probability of transitioning to any other arbitrary state. All GUMDPs now become unichain. In Fig. 3b, we display the results obtained for different $\gamma$ values with trajectories of infinite length. As can be seen, for the discounted setting ($\gamma < 1$), it continues to exist a mismatch between the objectives. However, under the average setting ($\gamma \to 1$), the gap between the objectives fades away for all GUMDPs. Such results are in line with Theo. 4.4, where we showed that the different objectives are equivalent for unichain GUMDPs.

## 5. Related Work

To the authors' knowledge, Derman (1970, Chap. 7) was the first to highlight the mismatch between objectives that depend on empirical average occupancies in comparison to objectives that depend on expected average occupancies. The author notes that minimizing an objective function that depends on empirical average occupancies is distinct from minimizing an objective function that depends on expected average occupancies. However, the author does not provide concrete examples of this mismatch, nor characterizes the mismatch depending on different properties of the underlying GUMDP and the number of sampled trajectories.

Other works consider the case of empirical occupancies instead of expected occupancies. As an example, Ross & Varadarajan (1989; 1987; 1991); Haviv (1996) study average MDPs with sample-path constraints where the empirical cost induced by a trajectory needs to be below a certain threshold with probability one. Other works further characterize the convergence of empirical average occupancies in MDPs (Altman & Zeitouni, 1994; Mannor & Tsitsiklis, 2005; Tracol, 2009).

More recently, Mutti et al. (2023) show that finite-horizon GUMDPs implicitly make an infinite trials assumption, i.e., finite-horizon GUMDPs implicitly assume the performance of a given policy is evaluated under an infinite number of episodes of interaction with the environment. The authors introduce a modification of GUMDPs where the objective

Table 1: Are the infinite and finite trials formulations equivalent? (✓ = yes, ✗ = no)

| Objective function ($f$) | Discounted setting | Average setting | |
|---|---|---|---|
| | | **Unichain** | **Multichain** |
| **Linear** | ✓ [Remark 3.1] | ✓ [Remark 3.1] | ✓ [Remark 3.1] |
| **Non-linear** | ✗ [Theo. 4.1] | ✓ [Theo. 4.4] | ✗ [Theo. 4.5] |

function depends on the empirical state-action occupancy induced over a finite number of episodes. Under the introduced finite trials formulation, the authors show that the class of Markovian policies does not suffice, in general, to achieve optimality and that non-Markovian policies may need to be considered.

In our work, we extend the work developed by (Mutti et al., 2023) by considering the infinite-horizon setting. We start by showing that, under the infinite-horizon, there still exists, in general, a mismatch between the finite trials objective and the infinite trials objective. After establishing the existence of the mismatch for both the discounted and average settings, we further provide upper and lower bounds that allow to better understand how such a mismatch depends on the number of sampled trajectories. We also elaborate on how our bounds and the mismatch between the finite and infinite trials formulations depend on certain properties of the underlying GUMDP. We are the first work to provide lower bounds on the mismatch between the finite and infinite trials objectives, as Mutti et al. (2023) only prove upper bounds in the context of policy optimization. Also, while Mutti et al. (2023) show that, in general, there exists a mismatch between the finite and infinite trials formulations under finite-horizon settings, our Theo. 4.4 states that under certain conditions (the GUMDP being unichain), there exists no mismatch. Hence, the nature of our results and those in Mutti et al. (2023) greatly differ.

## 6. Conclusion

In this work, we provided clear evidence, both theoretically and empirically, that *the number of trials matters in infinite-horizon GUMDPs*. First, under the discounted setting, we showed that a mismatch between the finite and infinite trials formulations exists in general. We also provided upper and lower bounds to quantify such mismatch as a function of the number of sampled trajectories. Second, under the average setting, we showed how the structure of the underlying GUMDP influences the mismatch between the finite and infinite trials formulations: (i) for unichain GUMDPs, the infinite and finite trials formulations are equivalent; and (ii) for multichain GUMDPs there is, in general, a mismatch between the different objectives. Finally, we provided a set of empirical results to support our theoretical claims. We

summarize our results in Table 1.

While we focused on the case of policy evaluation, we expect the mismatch between the finite and infinite trials to also impact policy optimization. For example, under a generalized policy iteration scheme (Sutton & Barto, 2018), we expect the mismatch between the infinite and finite trials formulations at the policy evaluation stages to impact the resulting optimal policies. Future work should study policy optimization in the finite trials regime.

## Acknowledgements

This work was supported by Portuguese national funds through the Portuguese Fundação para a Ciência e a Tecnologia (FCT) under projects UIDB/50021/2020 (INESC-ID multi-annual funding), PTDC/CCI-COM/5060/2021 (RELEvaNT), and AI-PackBot (project number 14935, LISBOA2030-FEDER-00854700). Pedro P. Santos acknowledges the FCT PhD grant 2021.04684.BD.

## Impact Statement

This paper presents work whose goal is to advance the field of Machine Learning. There are many potential societal consequences of our work, none of which we feel must be specifically highlighted here.

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

# A. Empirical State-Action Occupancies

Given a random trajectory $T = (S_0, A_0, S_1, A_1, \ldots)$, we note that

$$\mathbb{P}_\pi(S_t = s, A_t = a | S_0 \sim p_0) = \mathbb{E}_{T \sim \zeta_\pi} \left[ \mathbf{1}(S_t = s, A_t = a) \right] = \sum_\tau \zeta_\pi(\tau) \mathbf{1}(s_t = s, a_t = a),$$

where $\zeta_\pi(\tau)$ denotes the probability of trajectory $\tau = (s_0, a_0, s_1, a_1, \ldots)$ under policy $\pi$.

## A.1. Discounted occupancies

Given a random trajectory $T = (S_0, A_0, S_1, A_1, \ldots)$, consider the estimator $\hat{d}_T(s, a)$ defined as

$$\hat{d}_T(s, a) = (1 - \gamma) \sum_{t=0}^{\infty} \gamma^t \mathbf{1}(S_t = s, A_t = a). \tag{8}$$

It holds that, for all $s \in \mathcal{S}$ and $a \in \mathcal{A}$,

$$
\begin{aligned}
\mathbb{E}_{T \sim \zeta_\pi} \left[ \hat{d}_T(s, a) \right] &= \mathbb{E}_{T \sim \zeta_\pi} \left[ (1 - \gamma) \sum_{t=0}^{\infty} \gamma^t \mathbf{1}(S_t = s, A_t = a) \right] \\
&= (1 - \gamma) \sum_\tau \zeta_\pi(\tau) \sum_{t=0}^{\infty} \gamma^t \mathbf{1}(s_t = s, a_t = a) \\
&= (1 - \gamma) \sum_{t=0}^{\infty} \gamma^t \sum_\tau \zeta_\pi(\tau) \mathbf{1}(s_t = s, a_t = a) \\
&= (1 - \gamma) \sum_{t=0}^{\infty} \gamma^t \mathbb{P}_\pi(S_t = s, A_t = a | s_0 \sim p_0) \\
&= d_\pi(s, a),
\end{aligned}
$$

i.e., $\hat{d}_T$ is an unbiased estimator for $d_\pi$. Given a set of $K$ random trajectories $\mathcal{T}_K = \{T_1, \ldots, T_K\}$, consider estimator

$$\hat{d}_{\mathcal{T}_K}(s, a) = \frac{1}{K} \sum_{k=1}^{K} \hat{d}_{T_k}(s, a),$$

for $\hat{d}_{T_k}$ as defined in (8). We note again that, for all $s \in \mathcal{S}$ and $a \in \mathcal{A}$, and any $K \in \mathbb{N}$,

$$
\begin{aligned}
\mathbb{E}_{\mathcal{T}_K} \left[ \hat{d}_{\mathcal{T}_K}(s, a) \right] &= \mathbb{E}_{\mathcal{T}_K} \left[ \frac{1}{K} \sum_{k=1}^{K} \hat{d}_{T_k}(s, a) \right] \\
&= \frac{1}{K} \sum_{k=1}^{K} \mathbb{E}_{T_k \sim \zeta_\pi} \left[ \hat{d}_{T_k}(s, a) \right] \\
&= d_\pi(s, a),
\end{aligned}
$$

i.e., the estimator $\hat{d}_{\mathcal{T}_K}$ is unbiased. We now show that estimator $\hat{d}_{\mathcal{T}_K}$ is also consistent.

*Remark* A.1. ($\hat{d}_{\mathcal{T}_K}$ is a consistent estimator) For any $s \in \mathcal{S}$ and $a \in \mathcal{A}$, the estimator $\hat{d}_{\mathcal{T}_K}(s, a)$ is consistent in probability for $d_\pi(s, a)$, i.e., $\lim_{K \to \infty} \mathbb{P} \left( \left| \hat{d}_{\mathcal{T}_K}(s, a) - d_\pi(s, a) \right| > \epsilon \right) = 0, \forall \epsilon > 0$. This is true because the estimator consists of a sample average of random variables $\tilde{d}_k(s, a) = \hat{d}_{T_k}(s, a)$ (we note that $\hat{d}_{T_k}(s, a)$ is a random variable since it is the result of applying a function to the random trajectory $T_k$). In particular, since random variables $\tilde{d}_k(s, a)$ are i.i.d. and $\mathbb{E} \left[ \tilde{d}_k(s, a) \right] = d_\pi(s, a) < \infty$, for all $k$, the weak law of large numbers states that $\frac{1}{K} \sum_{k=1}^{K} \tilde{d}_k(s, a)$ converges in probability to $d_\pi(s, a)$ when $K \to \infty$.

### A.2. Average occupancies

Given a random trajectory $T = (S_0, A_0, S_1, A_1, \ldots)$, consider estimator $\hat{d}_T(s, a)$ defined as

$$\hat{d}_T(s, a) = \lim_{H \to \infty} \frac{1}{H} \sum_{t=0}^{H-1} \mathbf{1}(S_t = s, A_t = a). \tag{9}$$

It holds that, for all $s \in \mathcal{S}$ and $a \in \mathcal{A}$,

$$\begin{aligned}
\mathbb{E}_{T \sim \zeta_\pi}\left[\hat{d}_T(s, a)\right] &= \mathbb{E}_{T \sim \zeta_\pi}\left[\lim_{H \to \infty} \frac{1}{H} \sum_{t=0}^{H-1} \mathbf{1}(S_t = s, A_t = a)\right] \\
&= \sum_\tau \zeta_\pi(\tau) \lim_{H \to \infty} \frac{1}{H} \sum_{t=0}^{H-1} \mathbf{1}(S_t = s, A_t = a) \\
&= \lim_{H \to \infty} \frac{1}{H} \sum_\tau \zeta_\pi(\tau) \sum_{t=0}^{H-1} \mathbf{1}(S_t = s, A_t = a) \\
&= \lim_{H \to \infty} \frac{1}{H} \sum_{t=0}^{H-1} \sum_\tau \zeta_\pi(\tau) \mathbf{1}(S_t = s, A_t = a) \\
&= \lim_{H \to \infty} \frac{1}{H} \sum_{t=0}^{H-1} \mathbb{P}_\pi(S_t = s, A_t = a) \\
&= d_{\mathrm{avg}, \pi}(s, a),
\end{aligned}$$

i.e., $\hat{d}_T$ is an unbiased estimator for $d_{\mathrm{avg}, \pi}$. Given a set of $K$ random trajectories $\mathcal{T}_K = \{T_1, \ldots, T_K\}$, consider estimator

$$\hat{d}_{\mathcal{T}_K}(s, a) = \frac{1}{K} \sum_{k=1}^{K} \hat{d}_{T_k}(s, a),$$

for $\hat{d}_{T_k}$ as defined in (9). We note again that, for all $s \in \mathcal{S}$ and $a \in \mathcal{A}$, and any $K \in \mathbb{N}$,

$$\begin{aligned}
\mathbb{E}_{\mathcal{T}_K}\left[\hat{d}_{\mathcal{T}_K}(s, a)\right] &= \mathbb{E}_{\mathcal{T}_K}\left[\frac{1}{K} \sum_{k=1}^{K} \hat{d}_{T_k}(s, a)\right] \\
&= \frac{1}{K} \sum_{k=1}^{K} \mathbb{E}_{T_k \sim \zeta_\pi}\left[\hat{d}_{T_k}(s, a)\right] \\
&= d_{\mathrm{avg}, \pi}(s, a),
\end{aligned}$$

i.e., the estimator $\hat{d}_{\mathcal{T}_K}$ is unbiased. Finally, similarly to the case of discounted occupancies, the average occupancy estimator $\hat{d}_{\mathcal{T}_K}$ is also consistent, i.e., $\lim_{K \to \infty} \mathbb{P}\left(\left|\hat{d}_{\mathcal{T}_K}(s, a) - d_\pi(s, a)\right| > \epsilon\right) = 0, \forall \epsilon > 0$. The line of reasoning is the same as that in Remark A.1.

## B. Policy Evaluation in the Finite Trials Regime

**Assumption B.1.** We say that $f$ is $c$-strongly convex if there exists $c > 0$ such that

$$f(d_1) \geq f(d_2) + \nabla f(d_2)^\top (d_1 - d_2) + \frac{c}{2} \|d_1 - d_2\|_2^2, \tag{10}$$

for any $d_1, d_2$ belonging to the domain of $f$. Equivalently, $f$ is $c$-strongly convex if there exists $c > 0$ such that

$$\nabla^2 f(d) \succeq cI, \tag{11}$$

for all $d$ belonging to the domain of $f$, where $I$ is the identity matrix.

*Remark* B.2. Consider the objective function $f(d) = \langle d, \log(d) \rangle$. It holds that $f$ is $c$-strongly convex and any $c \in (0, 1]$ satisfies (10) and (11).

*Proof.* It holds that $\nabla^2 f(d) = \text{diag}([1/d(1), \ 1/d(2), \ \dots])$ where $\text{diag}(a)$ denotes the diagonal matrix with vector $a$ in its diagonal. From the strong convexity definition,

$$\nabla^2 f(d) \succeq cI, \ \forall d$$
$$\iff \nabla^2 f(d) - cI \succeq 0, \ \forall d$$
$$\iff \lambda_i(d) - c \geq 0, \ \forall i, \ \forall d$$
$$\iff \lambda_{\min}(d) \geq c, \ \forall d$$
$$\iff \min_i \frac{1}{d(i)} \geq c, \ \forall d,$$

where $\lambda_i(d)$ are the eigenvalues of matrix $\nabla^2 f(d)$. Since $d \in \Delta(\mathcal{S} \times \mathcal{A})$, it holds that $\min_i \frac{1}{d(i)} \geq 1$ for any $d \in \Delta(\mathcal{S} \times \mathcal{A})$ and, hence, $f$ is $c$-strongly convex with any $c \in (0, 1]$ satisfying (10) and (11). $\qquad\square$

*Remark* B.3. Consider the objective function $f(d) = \text{KL}(d|d_\beta) = \sum_i d(i) \log\left(\frac{d(i)}{d_\beta(i)}\right)$, where $d_\beta$ is fixed. It holds that $f$ is $c$-strongly convex and any $c \in (0, 1]$ satisfies (10) and (11).

*Proof.* It holds that $\nabla^2 f(d) = \text{diag}([1/d(1), \ 1/d(2), \ \dots])$ where $\text{diag}(a)$ denotes the diagonal matrix with vector $a$ in its diagonal. Thus, similar to the case of the entropy function, it holds that $f$ is $c$-strongly convex with any $c \in (0, 1]$ satisfying (10) and (11). $\qquad\square$

*Remark* B.4. Consider the objective function $f(d) = d^\top A d$. If $A$ is positive definite, i.e., $\lambda_{\min}(A) > 0$ where $\lambda_{\min}(A)$ denotes the smallest eigenvalue of matrix $A$, then $f$ is $c$-strongly convex. Furthermore, any $c \in (0, 2\lambda_{\min}(A)]$ satisfies (10) and (11).

*Proof.* It holds that $\nabla^2 f(d) = 2A$. From the condition for strong convexity,

$$\nabla^2 f(d) \succeq cI, \ \forall d$$
$$\iff 2A - cI \succeq 0$$
$$\iff 2\lambda_i - c \geq 0, \ \forall i$$
$$\iff 2\lambda_i \geq c, \ \forall i$$
$$\iff 2\lambda_{\min}(A) \geq c,$$

where $\lambda_i$ are the eigenvalues of matrix $A$. If $\lambda_{\min}(A) > 0$, then any $c \in (0, 2\lambda_{\min}(A)]$ satisfies the condition above. $\qquad\square$

## B.1. Discounted setting

### B.1.1. PROOF OF THEOREM 4.2

*Theorem* 4.2. Let $\mathcal{M}_f$ be a GUMDP with $c$-strongly convex $f$ and $K \in \mathbb{N}$ be the number of sampled trajectories. Then, for any policy $\pi \in \Pi_S$ it holds that

$$f_K(\pi) - f_\infty(\pi) \geq \frac{c}{2K} \sum_{\substack{s \in \mathcal{S} \\ a \in \mathcal{A}}} \underset{T \sim \zeta_\pi}{\text{Var}} \left[\hat{d}_T(s, a)\right]$$

$$= \frac{c(1-\gamma)^2}{2K} \sum_{\substack{s \in \mathcal{S} \\ a \in \mathcal{A}}} \underset{T \sim \zeta_\pi}{\text{Var}} \left[J_{r_{s,a}}^{\gamma, \pi}\right],$$

where $J_{r_{s,a}}^{\gamma, \pi} = \sum_{t=0}^{\infty} \gamma^t r_{s,a}(S_t, A_t)$ is the discounted return for the MDP with reward function $r_{s,a}(s', a') = 1$ if $s' = s$ and $a' = a$, and zero otherwise.

*Proof.* For any policy $\pi \in \Pi_S$ it holds that

$$
\begin{aligned}
f_K(\pi) - f_\infty(\pi) &= \mathbb{E}_{\mathcal{T}_K}\left[f(\hat{d}_{\mathcal{T}_K})\right] - f\left(\mathbb{E}_{\mathcal{T}_K}\left[\hat{d}_{\mathcal{T}_K}\right]\right) \\
&\overset{(a)}{\geq} \frac{c}{2}\mathbb{E}_{\mathcal{T}_K}\left[\left\|\hat{d}_{\mathcal{T}_K} - d_\pi\right\|_2^2\right] \\
&= \frac{c}{2}\mathbb{E}_{\mathcal{T}_K}\left[\sum_{s\in\mathcal{S}, a\in\mathcal{A}}\left(\hat{d}_{\mathcal{T}_K}(s,a) - d_\pi(s,a)\right)^2\right] \\
&= \frac{c}{2}\sum_{s\in\mathcal{S}, a\in\mathcal{A}}\mathbb{E}_{\mathcal{T}_K}\left[\left(\hat{d}_{\mathcal{T}_K}(s,a) - d_\pi(s,a)\right)^2\right] \\
&= \frac{c}{2}\sum_{s\in\mathcal{S}, a\in\mathcal{A}}\mathrm{Var}_{\mathcal{T}_K}\left[\hat{d}_{\mathcal{T}_K}(s,a)\right] \\
&= \frac{c}{2}\sum_{s\in\mathcal{S}, a\in\mathcal{A}}\mathrm{Var}_{\{T_1,\ldots,T_K\}}\left[\frac{1}{K}\sum_{k=1}^{K}\hat{d}_{T_k}(s,a)\right] \\
&= \frac{c}{2K}\sum_{s\in\mathcal{S}, a\in\mathcal{A}}\mathrm{Var}_{T\sim\zeta_\pi}\left[\hat{d}_T(s,a)\right] \\
&= \frac{c(1-\gamma)^2}{2K}\sum_{s\in\mathcal{S}, a\in\mathcal{A}}\mathrm{Var}_{T\sim\zeta_\pi}\left[\sum_{t=0}^{\infty}\gamma^t\mathbf{1}(S_t = s, A_t = a)\right], \\
&= \frac{c(1-\gamma)^2}{2K}\sum_{s\in\mathcal{S}, a\in\mathcal{A}}\mathrm{Var}_{T\sim\zeta_\pi}\left[J_{r_{s,a}}^{\gamma,\pi}\right],
\end{aligned}
$$

where (a) follows from the strongly convex assumption and the fact that

$$
f(X) \geq f(\mathbb{E}[X]) + \nabla f(\mathbb{E}[X])^\top (X - \mathbb{E}[X]) + \frac{c}{2}\|X - \mathbb{E}[X]\|_2^2
$$
$$
\implies \mathbb{E}[f(X)] \geq f(\mathbb{E}[X]) + \frac{c}{2}\mathbb{E}\left[\|X - \mathbb{E}[X]\|_2^2\right],
$$

where $X$ is a random vector. We refer to (Benito, 1982; Sobel, 1982; Sitař, 2006) for a closed-form expression for the calculation of $\mathrm{Var}_{T\sim\zeta_\pi}\left[J_{r_{s,a}}^{\gamma,\pi}\right]$. $\qquad\square$

### B.1.2. PROOF OF THEOREM 4.3

*Theorem* 4.3. Let $\mathcal{M}_f$ be a GUMDP with convex and $L$-Lipschitz $f$, $K \in \mathbb{N}$ be the number of sampled trajectories, each with length $H \in \mathbb{N}$. Then, for any policy $\pi$ and $\delta \in (0,1]$ it holds with probability at least $1 - \delta$

$$
|f_\infty(\pi) - f(\hat{d}_{\mathcal{T}_K,H})| \leq L\left(\sqrt{\frac{2|\mathcal{S}||\mathcal{A}|\log(2H/\delta)}{K}} + 2\gamma^H\right) = E_{\text{Upper}}(K, H).
$$

For fixed $H \in \mathbb{N}$ it holds that

$$
\lim_{K\to\infty} E_{\text{Upper}}(K, H) = 2L\gamma^H,
$$

and for fixed $K \in \mathbb{N}$

$$
\lim_{H\to\infty} E_{\text{Upper}}(K, H) = \infty.
$$

*Proof.* For any policy $\pi$ and $H \in \mathbb{N}$, we have

$$
\left| f_\infty(\pi) - f(\hat{d}_{\mathcal{T}_K, H}) \right| = \left| f(d_\pi) - f(\hat{d}_{\mathcal{T}_K, H}) \right|
$$

$$
\overset{(a)}{\leq} L \left\| d_\pi - \hat{d}_{\mathcal{T}_K, H} \right\|_1
$$

$$
\overset{(b)}{=} L \left\| (1-\gamma) \sum_{t=0}^\infty \gamma^t d_{\pi, t} - \frac{(1-\gamma)}{1-\gamma^H} \sum_{t=0}^{H-1} \gamma^t \hat{d}_{K,t} \right\|_1
$$

$$
= L \left\| \frac{1-\gamma}{1-\gamma^H} \sum_{t=0}^{H-1} \gamma^t \left( (1-\gamma^H) d_{\pi,t} - \hat{d}_{K,t} \right) + (1-\gamma) \sum_{t=H}^\infty \gamma^t d_{\pi,t} \right\|_1
$$

$$
\overset{(c)}{\leq} L \left\| \frac{1-\gamma}{1-\gamma^H} \sum_{t=0}^{H-1} \gamma^t \left( (1-\gamma^H) d_{\pi,t} - \hat{d}_{K,t} \right) \right\|_1 + \gamma^H
$$

$$
\overset{(d)}{\leq} L \left( \frac{1-\gamma}{1-\gamma^H} \sum_{t=0}^{H-1} \gamma^t \left\| (1-\gamma^H) d_{\pi,t} - \hat{d}_{K,t} \right\|_1 + \gamma^H \right)
$$

$$
\overset{(e)}{\leq} L \left( \frac{1-\gamma}{1-\gamma^H} \sum_{t=0}^{H-1} \gamma^t \left( \left\| \hat{d}_{K,t} - d_{\pi,t} \right\|_1 + \left\| \gamma^H d_{\pi,t} \right\|_1 \right) + \gamma^H \right)
$$

$$
\leq L \left( \frac{1-\gamma}{1-\gamma^H} \sum_{t=0}^{H-1} \gamma^t \left\| \hat{d}_{K,t} - d_{\pi,t} \right\|_1 + 2\gamma^H \right)
$$

$$
\leq L \left( \max_{t \in \{0, \ldots, H-1\}} \left\| \hat{d}_{K,t} - d_{\pi,t} \right\|_1 + 2\gamma^H \right)
$$

where: (a) is due to the $L$-Lipschitz assumption; in (b) we used $d_\pi = (1-\gamma) \sum_{t=0}^\infty \gamma^t d_{\pi,t}$ where $d_{\pi,t}$ denotes the expected occupancy under policy $\pi$ at timestep $t$, and $\hat{d}_{\mathcal{T}_K, H} = (1-\gamma)/(1-\gamma^H) \sum_{t=0}^{H-1} \gamma^t \hat{d}_{K,t}$ where $\hat{d}_{K,t}$ denotes the empirical distribution induced by the $K$ random trajectories at timestep $t$; and (c), (d) and (e) follow from the triangular inequality. We aim to bound the last inequality above with high probability; to do so, we note that

$$
\mathbb{P} \left( \max_{t \in \{0, \ldots, H-1\}} \left\| \hat{d}_{K,t} - d_{\pi,t} \right\|_1 > \epsilon' \right) \leq \mathbb{P} \left( \bigcup_{t=0}^{H-1} \left\| \hat{d}_{K,t} - d_{\pi,t} \right\|_1 > \epsilon' \right)
$$

$$
\overset{(a)}{\leq} \sum_{t=0}^{H-1} \mathbb{P} \left( \left\| \hat{d}_{K,t} - d_{\pi,t} \right\|_1 > \epsilon' \right)
$$

$$
\overset{(b)}{\leq} \sum_{t=0}^{H-1} 2 \exp \left( -\frac{1}{2|\mathcal{S}||\mathcal{A}|} K(\epsilon')^2 \right)
$$

$$
= 2H \exp \left( -\frac{1}{2|\mathcal{S}||\mathcal{A}|} K(\epsilon')^2 \right).
$$

where: (a) follows from a union bound, and (b) from the fact that $\mathbb{P} \left( \left\| d_{\pi,t} - \hat{d}_{K,t} \right\| > \epsilon' \right) \leq 2 \exp \left( -\frac{1}{2|\mathcal{S}||\mathcal{A}|} K(\epsilon')^2 \right)$ (Lemma 16 in (Efroni et al., 2020)). Thus, it holds with probability at least $1 - \delta$

$$
\max_{t \in \{0, \ldots, H-1\}} \left\| \hat{d}_{K,t} - d_{\pi,t} \right\|_1 \leq \sqrt{\frac{2|\mathcal{S}||\mathcal{A}| \log(2H/\delta)}{K}}.
$$

Given the above we conclude that, with probability at least $1 - \delta$,

$$
\left| f_\infty(\pi) - f(\hat{d}_{\mathcal{T}_K, H}) \right| \leq L \left( \sqrt{\frac{2|\mathcal{S}||\mathcal{A}| \log(2H/\delta)}{K}} + 2\gamma^H \right).
$$

For the limits we have, for fixed $H \in \mathbb{N}$,

$$\lim_{K \to \infty} E_{\text{Upper}}(K, H) = \lim_{K \to \infty} L \left( \sqrt{\frac{2|\mathcal{S}||\mathcal{A}| \log(2H/\delta)}{K}} + 2\gamma^H \right)$$

$$= L \lim_{K \to \infty} \sqrt{\frac{2|\mathcal{S}||\mathcal{A}| \log(2H/\delta)}{K}} + 2L\gamma^H$$

$$= 0 + 2L\gamma^H.$$

For fixed $K \in \mathbb{N}$,

$$\lim_{H \to \infty} E_{\text{Upper}}(K, H) = \lim_{H \to \infty} L \left( \sqrt{\frac{2|\mathcal{S}||\mathcal{A}| \log(2H/\delta)}{K}} + 2\gamma^H \right)$$

$$= L \lim_{H \to \infty} \sqrt{\frac{2|\mathcal{S}||\mathcal{A}| \log(2H/\delta)}{K}} + 2L \lim_{H \to \infty} \gamma^H$$

$$= L\sqrt{\frac{2|\mathcal{S}||\mathcal{A}|}{K}} \lim_{H \to \infty} \sqrt{\log(2H/\delta)} + 0$$

$$= L\sqrt{\frac{2|\mathcal{S}||\mathcal{A}|}{K}} \sqrt{\lim_{H \to \infty} \log(2H/\delta)} + 0$$

$$= L\sqrt{\frac{2|\mathcal{S}||\mathcal{A}|}{K}} \cdot \infty + 0$$

$$= \infty$$

$\square$

## B.2. Average setting

### B.2.1. PROOF OF THEOREM 4.4

*Theorem* 4.4. If the GUMDP $\mathcal{M}_f$ is unichain and $f$ is bounded and continuous in its domain, then $f_K(\pi) = f_\infty(\pi)$ for any $\pi \in \Pi_S$.

*Proof.* We start by focusing on the case of state-dependant occupancies and later generalize our result for the case of state-action-dependant occupancies.

For any $\pi \in \Pi_S$, in a unichain GUMDP, the Markov chain with transition matrix $P^\pi$ and initial states distribution $p_0$ contains a single recurrent class $\mathcal{R}$, and a possibly non-empty set of transient states $\mathcal{Z}$. Associated to the unique recurrent class is $\mu_\pi \in \Delta(\mathcal{S})$, the unique stationary distribution of the Markov chain, which satisfies $\mu_\pi(s) > 0$ for $s \in \mathcal{R}$ and $\mu_\pi(s) = 0$ for $s \in \mathcal{Z}$. Furthermore, the unique stationary distribution $\mu_\pi$ satisfies

$$\sum_{s' \in \mathcal{S}} P^\pi(s|s')\mu_\pi(s') = \mu_\pi(s), \quad \forall s \in \mathcal{S}$$

$$\sum_{s \in \mathcal{S}} \mu_\pi(s) = 1.$$

All aforementioned facts can be found in textbooks such as (Puterman, 2014).

Now, for a fixed policy $\pi \in \Pi_S$, consider the estimator

$$\hat{d}_T(s) = \lim_{H \to \infty} \frac{1}{H} \sum_{t=0}^{H-1} \mathbf{1}(S_t = s), \tag{12}$$

where $T = (S_0, S_1, \ldots)$ denotes a random trajectory from the Markov chain with transition matrix $P^\pi$ and initial distribution $p_0$. Since there is a single recurrent class, independent of the initial state distribution, a random trajectory drawn from the Markov chain will eventually get absorbed into the unique recurrent class in a finite number of steps. Hence:

- for any transient state $s \in \mathcal{Z}$ of the Markov chain, we have that $\hat{d}_T(s) = 0$ almost surely. From the definition of a transient state, it holds that $\sum_{t=0}^{\infty} \mathbf{1}(S_t = s) < \infty$ with probability one, yielding

$$\hat{d}_T(s) = \lim_{H \to \infty} \frac{1}{H} \sum_{t=0}^{H-1} \mathbf{1}(S_t = s) \leq \lim_{H \to \infty} \frac{1}{H} \sum_{t=0}^{\infty} \mathbf{1}(S_t = s) = 0.$$

- for any recurrent state $s \in \mathcal{R}$ belonging to the unique recurrent class of the Markov chain, we have that $\hat{d}_T(s) = \mu_\pi(s)$ almost surely. Recurrent classes within a larger Markov chain can be seen as independent Markov chains (Puterman, 2014). Let $P_{R \to R}$ denote the transition matrix for the states belonging to the recurrent class of the Markov chain. Then, since $P_{R \to R}$ is an irreducible Markov chain (any state is reachable from any other state), from the ergodic theorem for Markov chains (Levin et al., 2006) it holds that, for any initial distribution,

$$\mathbb{P}\left(\hat{d}_T(s) = \mu_\pi(s)\right) = \mathbb{P}\left(\lim_{H \to \infty} \frac{1}{H} \sum_{t=0}^{H-1} \mathbf{1}(S_t = s) = \mu_\pi(s)\right) = 1.$$

Now, we use the fact that estimator $\hat{d}_T(s)$ converges almost surely to $\mu_\pi(s)$ in order to show the equivalence between $f_K(\pi)$ and $f_\infty(\pi)$. For now, we assume $f$ is defined over state-dependant occupancies and later extend our analysis for the case of state-action-dependant occupancies. Let $Z_{H,k}$ be the random vector with components $Z_{H,k}(s) = \frac{1}{H} \sum_{t=0}^{H-1} \mathbf{1}(S_{k,t} = s)$. Since, for each $s \in \mathcal{S}$ and $k \in \{1, \dots, K\}$, it holds that $Z_{H,k}(s) \to \mu_\pi(s)$ almost surely, we have that random vector $Z_{H,k} \to \mu_\pi$ almost surely, for any $k \in \{1, \dots, K\}$, i.e.,

$$\mathbb{P}\left(\lim_{H \to \infty} Z_{H,k} = \mu_\pi\right) = 1, \quad \forall k \in \{1, \dots, K\}.$$

By the definition of the finite trials objective (considering a state-dependant occupancy and objective function), it holds that

$$f_K(\pi) = \mathbb{E}_{\mathcal{T}_K}\left[f(\hat{d}_{\mathcal{T}_K})\right] = \mathbb{E}_{\mathcal{T}_K}\left[f\left(\frac{1}{K} \sum_{k=1}^{K} \lim_{H \to \infty} Z_{H,k}\right)\right].$$

Since $Z_{H,k} \to \mu_\pi$ almost surely for any $k \in \{1, \dots, K\}$, it also holds that $\frac{1}{K} \sum_{k=1}^{K} Z_{H,k} \to \mu_\pi$ almost surely since the $K$ trajectories are independently sampled. Assuming $f$ is continuous, it holds that $f(\frac{1}{K} \sum_{k=1}^{K} Z_{H,k}) \to f(\mu_\pi)$ almost surely. Since $f$ is bounded in its domain by assumption it also implies that $\left|f\left(\frac{1}{K} \sum_{k=1}^{K} Z_{H,k}\right)\right|$ is bounded for any $H \in \mathbb{N}$. Thus, from the dominated/bounded convergence theorem (Durrett, 2019, Theo. 1.6.7.), it holds that

$$\mathbb{E}_{\mathcal{T}_K}\left[f\left(\frac{1}{K} \sum_{k=1}^{K} \lim_{H \to \infty} Z_{H,k}\right)\right] = \mathbb{E}\left[f(\mu_\pi)\right] = f(\mu_\pi),$$

where the second equality holds because $f(\frac{1}{K} \sum_{k=1}^{K} Z_{H,k})$ converges almost surely to $f(\mu_\pi)$, which is a non-random quantity.

Finally, it remains to show that the result above also holds for the case of state-action occupancies. Consider a second Markov chain, which we call the extended Markov chain, that has state space $\tilde{\mathcal{S}} = \mathcal{S} \times \mathcal{A}$[3], transition matrix $\tilde{P}(s', a'|s, a) = p(s'|s, a)\pi(a'|s')$, and $\tilde{p}_0(s, a) = p_0(s)\pi(a|s)$. This Markov chain encapsulates both the transition dynamics $p$ and the policy $\pi$ within the transition matrix $\tilde{P}$ and it should be clear that a random trajectory from the extended Markov chain $((S_0, A_0), (S_1, A_1), \dots)$ precisely describes a random sequence of state-action pairs when using $\pi \in \Pi_S$ to interact with the GUMDP. It holds that the extended Markov chain has a unique stationary distribution $\tilde{\mu}_\pi(s, a) = \mu_\pi(s)\pi(a|s)$. This is true

---

[3]We denote a state of the extended Markov chain with the tuple $(s, a)$. However, to make the notation simpler, we usually drop the parenthesis from the $(s, a)$ tuple, thus writing $p_0(s, a)$ instead of $p_0((s, a))$, $\tilde{P}(s', a'|s, a)$ instead of $\tilde{P}((s', a')|(s, a))$, $\mu_\pi(s, a)$ instead of $\mu_\pi((s, a))$, etc.

because $\tilde{\mu}_\pi$ is a stationary distribution for the extended Markov chain if it satisfies

$$\tilde{\mu}_\pi(s', a') = \sum_{s \in \mathcal{S}} \sum_{a \in \mathcal{A}} \tilde{P}(s', a'|s, a)\tilde{\mu}_\pi(s, a), \ \forall s', a' \tag{13}$$

$$1 = \sum_{s \in \mathcal{S}} \sum_{a \in \mathcal{A}} \tilde{\mu}_\pi(s, a). \tag{14}$$

Letting $\tilde{\mu}_\pi(s, a) = \mu_\pi(s)\pi(a|s)$ satisfies (13) since, when $\pi(a'|s') > 0$,

$$\mu_\pi(s')\pi(a'|s') = \sum_{s \in \mathcal{S}} \sum_{a \in \mathcal{A}} \pi(a|s)\tilde{P}(s', a'|s, a)\mu_\pi(s)$$

$$\iff \mu_\pi(s')\pi(a'|s') = \sum_{s \in \mathcal{S}} \sum_{a \in \mathcal{A}} \pi(a|s)\pi(a'|s')p(s'|s, a)\mu_\pi(s)$$

$$\iff \mu_\pi(s') = \sum_{s \in \mathcal{S}} \sum_{a \in \mathcal{A}} \pi(a|s)p(s'|s, a)\mu_\pi(s)$$

$$\iff \mu_\pi(s') = \sum_{s \in \mathcal{S}} P^\pi(s'|s)\mu_\pi(s),$$

and the last equality above holds since $\mu_\pi$ is the stationary distribution of the Markov chain with transition matrix $P^\pi$. If $\pi(a'|s') = 0$, then (13) also holds given that $\tilde{\mu}_\pi(s', a') = \mu_\pi(s')\pi(a'|s') = 0$. Equation (14) is also straightforwardly satisfied. It can also be seen from the equations above that the stationary distribution $\tilde{\mu}_\pi$ is unique. Assume the opposite, i.e., there exist multiple stationary distributions for the extended Markov chain. This would imply that it also exist multiple vectors satisfying the last equation above, which we know it is not possible because the Markov chain with transition matrix $P^\pi$ has a unique stationary distribution.

Consider estimator

$$\hat{d}_T(s, a) = \lim_{H \to \infty} \frac{1}{H} \sum_{t=0}^{H-1} \mathbf{1}(S_t = s, A_t = a), \tag{15}$$

where $T = (S_0, A_0, S_1, A_1, \ldots)$ denotes a random sequence of state-action pairs from the extended Markov chain. Since there is a single recurrent class (associated with the unique stationary distribution of the extended Markov chain), independent of the initial state distribution, a random trajectory drawn from the extended Markov chain will eventually get absorbed into the unique recurrent class in a finite number of steps. Hence:

- if $\pi(a|s) = 0$ for some state-action pair $(s, a)$, then $(s, a)$ is never visited in the extended Markov chain under any distribution of initial states and, hence, $\mathbb{P}\left(\hat{d}_T(s, a) = 0\right) = 1$.

- for any state-action pair $(s, a)$ such that $\pi(a|s) > 0$, if $(s, a)$ is transient in the extended Markov chain (equivalent to $s$ being transient in the Markov chain $P^\pi$), then $\mathbb{P}\left(\hat{d}_T(s, a) = 0\right) = 1$.

- for any state-action pair $(s, a)$ such that $\pi(a|s) > 0$, if $(s, a)$ is recurrent in the extended Markov chain (equivalent to $s$ being recurrent in the Markov chain $P^\pi$), then from the Ergodic theorem for Markov chains (Levin et al., 2006)

$$\mathbb{P}\left(\hat{d}_T(s, a) = \tilde{\mu}_\pi(s, a)\right) = \mathbb{P}\left(\lim_{H \to \infty} \frac{1}{H} \sum_{t=0}^{H-1} \mathbf{1}(S_t = s, A_t = a) = \tilde{\mu}_\pi(s, a)\right) = 1.$$

Thus, noting that $\tilde{\mu}_\pi(s, a) = \mu_\pi(s)\pi(a|s) = d_\pi(s, a)$ and following the same line of reasoning as we did for the case of state-dependant occupancies, we can use the fact that estimator $\hat{d}_T(s, a)$ converges almost surely to $d_\pi(s, a)$ to show the equivalence between $f_K(\pi)$ and $f_\infty(\pi)$. $\qquad \square$

**Lemma B.5.** *Consider a Markov chain with finite state-space $\mathcal{S}$ and transition matrix $P$. Let $p_0 \in \Delta(\mathcal{S})$ be the distribution of initial states of the Markov chain, i.e., $S_0 \sim p_0$, and afterwards $S_t \sim P(\cdot|S_{t-1})$, for all $t > 0$. Assume that the state-space can be partitioned into $L$ disjoint recurrent classes $\mathcal{R}_1, \ldots, \mathcal{R}_L$ and a set of transient states $\mathcal{Z}$. For each recurrent state*

$s \in \mathcal{R}_1 \cup \ldots \cup \mathcal{R}_L$, we let $l(s)$ denote the index of the recurrent class to which state $s$ belongs. Let also $\mu_l$ denote the unique stationary distribution associated with each recurrent class $\mathcal{R}_l$. It holds, for any $s \in \mathcal{S}$, that

$$\mathbb{P}\left(\lim_{H \to \infty} \frac{1}{H} \sum_{t=0}^{H-1} \mathbf{1}(S_t = s) = Y_s\right) = 1,$$

where $Y_s$ is a random variable such that, if $s \in \mathcal{Z}$ then $\mathbb{P}(Y_s = 0) = 1$, and if $s$ is recurrent then

$$\mathbb{P}(Y_s = y) = \begin{cases} \alpha_{l(s)}, & \text{if } y = \mu_{l(s)}(s), \\ 1 - \alpha_{l(s)}, & \text{if } y = 0, \\ 0, & \text{otherwise}, \end{cases}$$

and

$$\alpha_{l(s)} = \lim_{t \to \infty} \mathbb{P}(S_t \in \mathcal{R}_{l(s)} | S_0 \sim p_0)$$

denotes the probability of absorption into recurrent class $\mathcal{R}_{l(s)}$, i.e., the recurrent class to which recurrent state $s$ belongs, when the initial state $S_0$ is distributed according to $p_0$ and can be calculated in a closed-form manner (*Kallenberg, 1983, Theo. 2.3.4.*). In other words, $\lim_{H \to \infty} \frac{1}{H} \sum_{t=0}^{H-1} \mathbf{1}(S_t = s)$ converges almost surely to random variable $Y_s$, which describes the asymptotic proportion of time the chain spends in any state $s \in \mathcal{S}$.

*Proof.* The state-space $\mathcal{S}$ of every finite-state Markov chain can be partitioned into $L$ disjoint recurrent classes $\mathcal{R}_1, \ldots, \mathcal{R}_L$ and a set of transient states $\mathcal{Z}$. For each recurrent state $s \in \mathcal{R}_1 \cup \ldots \cup \mathcal{R}_L$, we let $l(s)$ denote the index of the recurrent class to which state $s$ belongs. Every recurrent class $\mathcal{R}_l$ can be treated as an independent Markov chain, associated with a unique stationary distribution $\mu_l$, which satisfies $\mu_l(s) > 0$ for all $s \in \mathcal{R}_l$ and $\mu_l(s) = 0$ for all $s \notin \mathcal{R}_l$ (Puterman, 2014). Once absorbed into a given recurrent class, the chain cannot leave the recurrent class and will visit every state in the recurrent class infinitely often.

Let $\hat{d}_T(s) = \lim_{H \to \infty} \frac{1}{H} \sum_{t=0}^{H-1} \mathbf{1}(S_t = s)$ denote the asymptotic proportion of time the chain spends in any state $s \in \mathcal{S}$. It holds that:

- for any transient state $s \in \mathcal{Z}$ of the Markov chain, we have that $\hat{d}_T(s) = 0$ almost surely. From the definition of a transient state, it holds that $\sum_{t=0}^{\infty} \mathbf{1}(S_t = s) < \infty$ with probability one, yielding

$$\hat{d}_T(s) = \lim_{H \to \infty} \frac{1}{H} \sum_{t=0}^{H-1} \mathbf{1}(S_t = s) \leq \lim_{H \to \infty} \frac{1}{H} \sum_{t=0}^{\infty} \mathbf{1}(S_t = s) = 0.$$

- for each recurrent class $\mathcal{R}_l$, if the Markov chain gets absorbed into $\mathcal{R}_l$, then it holds that

$$\mathbb{P}\left(\lim_{H \to \infty} \frac{1}{H} \sum_{t=0}^{H-1} \mathbf{1}(S_t = s) = \mu_l(s)\right) = 1, \quad \text{for } s \in \mathcal{R}_l,$$

  i.e., the asymptotic proportion of time the chain spends in each state $s \in \mathcal{R}_l$ converges almost surely to $\mu_l(s)$ for all $s \in \mathcal{R}_l$. This result follows from the ergodic theorem for Markov chains (Levin et al., 2006).

- for each recurrent class $\mathcal{R}_l$, if the Markov chain gets absorbed into another recurrent class $l' \neq l$ then it holds that $\hat{d}_T(s) = 0$ for all $s \in \mathcal{R}_l$ since, once absorbed into class $\mathcal{R}_{l'}$ the chain cannot leave the recurrent class (by definition).

Therefore, the chain starts in an arbitrary state $S_0 \sim p_0$ and eventually gets absorbed with probability one into a given recurrent class $\mathcal{R}_l$. Once absorbed into $\mathcal{R}_l$, the chain behaves as an independent Markov chain defined only on states $\mathcal{R}_l$ and: (i) the chain cannot leave the recurrent class, hence $\hat{d}_T(s) = 0$ for all $s \notin \mathcal{R}_l$; and (ii) $\hat{d}_T(s)$ converges almost surely to $\mu_l(s)$ for every $s \in \mathcal{R}_l$. Thus, it holds that $\lim_{H \to \infty} \frac{1}{H} \sum_{t=0}^{H-1} \mathbf{1}(S_t = s)$ can be described, in the almost surely sense, by a random variable $Y_s$ such that: (i) if $s \in \mathcal{Z}$ then $Y_s = 0$ almost surely; (ii) if $s \in \mathcal{R}_l$, i.e., if $s$ belongs to some recurrent class $\mathcal{R}_l$, and the chain gets absorbed into $\mathcal{R}_l$, then $Y_s = \mu_l(s)$ almost surely; and (iii) if $s \in \mathcal{R}_l$ but the chain gets absorbed into

other recurrent class $\mathcal{R}_{l'}$, then $Y_s = 0$. Therefore, the probability density function of $Y_s$ can be described as follows: if $s \in \mathcal{Z}$, i.e., if $s$ is transient, then $\mathbb{P}\left(Y_s = 0\right) = 1$. On the other hand, if $s$ is recurrent

$$
\mathbb{P}\left(Y_s = y\right) = \begin{cases} \alpha_{l(s)}, & \text{if } y = \mu_{l(s)}(s), \\ 1 - \alpha_{l(s)}, & y = 0, \\ 0, & \text{otherwise,} \end{cases}
$$

where

$$
\alpha_{l(s)} = \lim_{t \to \infty} \mathbb{P}(S_t \in \mathcal{R}_{l(s)} | S_0 \sim p_0)
$$

is the probability of absorption to recurrent class $\mathcal{R}_{l(s)}$, i.e., the recurrent class to which recurrent state $s$ belongs, when $S_0 \sim p_0$ and can be calculated in a closed-form manner (Kallenberg, 1983, Theo. 2.3.4.) $\qquad\square$

**Lemma B.6.** *Consider a GUMDP with finite state-space $\mathcal{S}$, finite action-space $\mathcal{A}$, transition probability function $p : \mathcal{S} \times \mathcal{A} \to \Delta(\mathcal{S})$, and let $p_0 \in \Delta(\mathcal{S})$ be the distribution of initial states. For any fixed stationary policy $\pi \in \Pi_S$, the interaction between the policy and the GUMDP gives rise to a random sequence of state-action pairs $S_0, A_0, S_1, A_1, \ldots$. Consider also the Markov chain with state space $\mathcal{S}$, transition matrix $P^\pi(s'|s) = \sum_{a \in \mathcal{A}} p(s'|s, a)\pi(a|s)$, and initial distribution $p_0$. It holds, for any $s \in \mathcal{S}$ and $a \in \mathcal{A}$, that*

$$
\mathbb{P}\left(\lim_{H \to \infty} \frac{1}{H} \sum_{t=0}^{H-1} \mathbf{1}(S_t = s, A_t = a) = Y_s \pi(a|s)\right) = 1,
$$

*where $Y_s$ is the random variable defined in Lemma B.5 by considering the Markov chain with transition matrix $P^\pi$.*

*Proof.* For a given fixed policy $\pi \in \Pi_S$, consider two Markov chains:

- the first Markov chain has state space $\mathcal{S}$, transition matrix $P^\pi(s'|s) = \sum_{a \in \mathcal{A}} p(s'|s, a)\pi(a|s)$ and $p_0$ as defined in the original GUMDP. Assume this Markov chain can be partitioned into $L$ disjoint recurrent classes $\mathcal{R}_1, \ldots, \mathcal{R}_L$ and a set of transient states $\mathcal{Z}$. For each recurrent state $s \in \mathcal{R}_1 \cup \ldots \cup \mathcal{R}_L$, we let $l(s)$ denote the index of the recurrent class to which state $s$ belongs. Let $\alpha_l$ denote the probability of absorption to recurrent class $\mathcal{R}_l$ given $p_0$. Every recurrent class $\mathcal{R}_l$ can be treated as an independent Markov chain, associated with a unique stationary distribution $\mu_l$, which satisfies $\mu_l(s) > 0$ for all $s \in \mathcal{R}_l$ and $\mu_l(s) = 0$ for all $s \notin \mathcal{R}_l$ (Puterman, 2014).

- the second Markov chain, which we call extended Markov chain, has state space $\tilde{\mathcal{S}} = \mathcal{S} \times \mathcal{A}$ (hence we denote with the pair $(s, a)$ a given state of the extended Markov chain), transition matrix $\tilde{P}(s', a'|s, a) = p(s'|s, a)\pi(a'|s')$, and $\tilde{p}_0(s, a) = p_0(s)\pi(a|s)$. This Markov chain encapsulates both the transition dynamics $p$ and the policy $\pi$ within the transition matrix $\tilde{P}$. It should also be clear that a random trajectory from the extended Markov chain $((S_0, A_0), (S_1, A_1), \ldots)$ precisely describes a random sequence of state-action pairs when using $\pi \in \Pi_S$ to interact with the GUMDP. Assume we can partition the extended Markov chain into $L$ disjoint recurrent classes $\tilde{\mathcal{R}}_1, \ldots, \tilde{\mathcal{R}}_L$ and a set of transient states $\tilde{\mathcal{Z}}$. For each recurrent state $(s, a) \in \tilde{\mathcal{R}}_1 \cup \ldots \cup \tilde{\mathcal{R}}_L$, we let $\tilde{l}(s, a)$ denote the index of the recurrent class to which state $(s, a)$ belongs. Let also $\tilde{\alpha}_l$ denote the probability of absorption to recurrent class $\tilde{\mathcal{R}}_l$. Every recurrent class $\tilde{\mathcal{R}}_l$ can be treated as an independent Markov chain, associated with a unique stationary distribution $\tilde{\mu}_l$, which satisfies $\tilde{\mu}_l(s, a) > 0$ for all $s \in \tilde{\mathcal{R}}_l$ and $\mu_l(s, a) = 0$ for all $s \notin \tilde{\mathcal{R}}_l$ (Puterman, 2014).

We are now interested in understanding how the long-term behavior of both chains is related, for fixed $\pi \in \Pi_S$. We make the following remarks:

1. if $\pi(a|s) = 0$ for some state-action pair $(s, a)$, then $(s, a)$ is never visited in the extended Markov chain for any distribution of initial states $p_0$.

2. For each $(s, a)$ such that $\pi(a|s) > 0$, if $(s, a)$ is transient, then all states $(s, a')$ for $a' \in \mathcal{A}$ such that $\pi(a'|s) > 0$ are also transient. This is true because if some pair $(s, a)$ is only finitely visited (from the definition of a transient state) it implies that state $s$ is also finitely visited and, therefore, all pairs $(s, a')$ for $a' \in \mathcal{A}$ such that $\pi(a'|s) > 0$ need also to be finitely visited. For each $(s, a)$ such that $\pi(a|s) > 0$, if $(s, a)$ is recurrent, then all states $(s, a')$ for $a' \in \mathcal{A}$ such that $\pi(a'|s) > 0$ are also recurrent. This is true because, if some pair $(s, a)$ is infinitely visited (from the definition of a recurrent state) it implies that state $s$ is also infinitely visited and, therefore, all pairs $(s, a')$ for $a' \in \mathcal{A}$ such that $\pi(a'|s) > 0$ need also to be infinitely visited.

3. A given state $s$ is transient for $P^\pi$ if and only if states $\{(s,a) : \pi(a|s) > 0\}$ are transient for $\tilde{P}$. A given state $s$ is recurrent for $P^\pi$ if and only if states $\{(s,a) : \pi(a|s) > 0\}$ are recurrent for $\tilde{P}$.

4. Two states $s$ and $s'$ are reachable from each other for $P^\pi$ if and only if all states in $\{(s,a) : \pi(a|s) > 0\}$ and all states in $\{(s',a) : \pi(a|s') > 0\}$ are reachable from each other for $\tilde{P}$.

5. Given the point above, a given set of states $\mathcal{C}$ is communicating for $P^\pi$, i.e., all states in $\mathcal{C}$ are reachable from each other according to $P^\pi$, if and only if the set of states $\{(s,a) : s \in \mathcal{C} \text{ and } \pi(a|s) > 0\}$ is communicating for $\tilde{P}$.

6. A communicating class $\mathcal{C}$ is closed for $P^\pi$, i.e., it is impossible to leave $\mathcal{C}$, if and only if the set of states $\{(s,a) : s \in \mathcal{C} \text{ and } \pi(a|s) > 0\}$ is closed for $\tilde{P}$.

7. Since a recurrent class is a set of states that is communicating and closed, given the two points above, there exists a one-to-one correspondence between the recurrent classes $\mathcal{R}_1, \ldots, \mathcal{R}_L$ and $\tilde{\mathcal{R}}_1, \ldots, \tilde{\mathcal{R}}_L$. In particular, $\tilde{\mathcal{R}}_l = \{(s,a) : s \in \mathcal{R}_l \text{ and } \pi(a|s) > 0\}$ and $\mathcal{R}_l = \{s : (s,a) \in \tilde{R}_l \text{ and } \pi(a|s) > 0\}$.

8. The probabilities of absorption into any of the recurrent classes are the same for both Markov chains, i.e., $\alpha_l = \tilde{\alpha}_l$ for all $l \in \{1, \ldots, L\}$. Let $\alpha_{s,l}$ denote the probability of absorption into recurrent class $\mathcal{R}_l$ in the Markov chain $P^\pi$ when the initial state is $s$, and $\tilde{\alpha}_{(s,a),l}$ denote the probability of absorption into recurrent class $\tilde{\mathcal{R}}_l$ in the extended Markov chain when the initial state is $(s,a)$. It holds, for any $l \in \{1, \ldots, L\}$, that

$$\alpha_l = \sum_{s \in \mathcal{S}} p_0(s) \alpha_{s,l} \tag{16}$$

and

$$\tilde{\alpha}_l = \sum_{s \in \mathcal{S}} \sum_{a \in \mathcal{A}} p_0(s,a) \tilde{\alpha}_{(s,a),l}$$
$$= \sum_{s \in \mathcal{S}} \sum_{a \in \mathcal{A}} p_0(s) \pi(a|s) \tilde{\alpha}_{(s,a),l}.$$

It should also be clear that $\alpha_{s,l} = \sum_{a \in \mathcal{A}} \pi(a|s) \tilde{\alpha}_{(s,a),l}$ given the one-to-one correspondence between the recurrent classes of both Markov chains. Replacing $\alpha_{s,l} = \sum_{a \in \mathcal{A}} \pi(a|s) \tilde{\alpha}_{(s,a),l}$ in (16) yields $\tilde{\alpha}_l$ and, hence, $\alpha_l = \tilde{\alpha}_l$. We refer to (Kallenberg, 1983, Theo. 2.3.4.) for a closed-form expression to calculate the probabilities of absorption into each recurrent class.

9. For every recurrent class $\tilde{\mathcal{R}}_l$ of the extended Markov chain it holds that $\tilde{\mu}_l(s,a) = \mu_l(s)\pi(a|s)$. This is true because $\tilde{\mu}_l$ is a stationary distribution for the extended Markov chain if it satisfies

$$\tilde{\mu}_l(s',a') = \sum_{s \in \mathcal{S}} \sum_{a \in \mathcal{A}} \tilde{P}(s',a'|s,a) \tilde{\mu}_l(s,a), \; \forall s', a' \tag{17}$$

$$1 = \sum_{s \in \mathcal{S}} \sum_{a \in \mathcal{A}} \tilde{\mu}_l(s,a). \tag{18}$$

Letting $\tilde{\mu}_l(s,a) = \mu_l(s)\pi(a|s)$ satisfies (17) since, when $\pi(a'|s') > 0$,

$$\mu_l(s')\pi(a'|s') = \sum_{s \in \mathcal{S}} \sum_{a \in \mathcal{A}} \pi(a|s) \tilde{P}(s',a'|s,a) \mu_l(s)$$

$$\iff \mu_l(s')\pi(a'|s') = \sum_{s \in \mathcal{S}} \sum_{a \in \mathcal{A}} \pi(a|s) \pi(a'|s') p(s'|s,a) \mu_l(s)$$

$$\iff \mu_l(s') = \sum_{s \in \mathcal{S}} \sum_{a \in \mathcal{A}} \pi(a|s) p(s'|s,a) \mu_l(s)$$

$$\iff \mu_l(s') = \sum_{s \in \mathcal{S}} P^\pi(s'|s) \mu_l(s),$$

and the last equality above holds since $\mu_l$ is the stationary distribution of the Markov chain with transition matrix $P^\pi$. If $\pi(a'|s') = 0$, then (17) also holds given that $\tilde{\mu}_l(s',a') = \mu_l(s')\pi(a'|s') = 0$. Equation (18) is also straightforwardly satisfied. It can also be seen from the equations above that the stationary distribution $\tilde{\mu}_\pi$ is unique.

In summary, the limiting behavior of the extended Markov chain can be described by the Markov chain with transition matrix $P^\pi$ since the recurrent classes of both Markov chains are intrinsically related (with the same probabilities of absorption), and the stationary distributions for each of the recurrent classes $\tilde{\mathcal{R}}_l$ of the extended Markov chain satisfy $\tilde{\mu}_l(s, a) = \mu_l(s)\pi(a|s)$.

Thus, for the extended Markov chain and any $(s, a)$, according to Lemma B.5, it holds that

$$\mathbb{P}\left(\lim_{H \to \infty} \frac{1}{H} \sum_{t=0}^{H-1} \mathbf{1}(S_t = s, A_t = a) = \tilde{Y}_{(s,a)}\right) = 1,$$

where $\tilde{Y}_{(s,a)}$ is a random variable such that, if $(s, a) \in \tilde{\mathcal{Z}}$ then $\mathbb{P}(\tilde{Y}_{(s,a)} = 0) = 1$, and if $(s, a)$ belongs to recurrent class $\tilde{\mathcal{R}}_l$

$$\mathbb{P}\left(\tilde{Y}_{(s,a)} = y\right) = \begin{cases} \tilde{\alpha}_l, & \text{if } y = \tilde{\mu}_l(s, a), \\ 1 - \tilde{\alpha}_l, & \text{if } y = 0, \\ 0, & \text{otherwise}, \end{cases}$$

and

$$\tilde{\alpha}_l = \lim_{t \to \infty} \mathbb{P}(\tilde{S}_t \in \tilde{\mathcal{R}}_l | \tilde{S}_0 \sim \tilde{p}_0)$$

denotes the probability of absortion into recurrent class $\tilde{\mathcal{R}}_l$ when the initial state $\tilde{S}_0$ is distributed according to $\tilde{p}_0$. Since there exists a one-to-one equivalence between the sets of recurrent classes of both Markov chains, the probabilities of absorption into each recurrent class are the same in both Markov chains, and $\tilde{\mu}_l(s, a) = \mu_l(s)\pi(a|s)$, it holds that random variable $\tilde{Y}_{(s,a)}$ can be equivalently described by $Y_s\pi(a|s)$, where $Y_s$ is the random variable defined in Lemma B.5 for the Markov chain with transition matrix $P^\pi$. $\qquad\square$

### B.2.2. PROOF OF THEOREM 4.6

*Theorem* 4.6. Let $\mathcal{M}_f$ be an average GUMDP with $c$-strongly convex $f$ and $K \in \mathbb{N}$ be the number of sampled trajectories. Consider also the Markov chain with state-space $\mathcal{S}$, transition matrix $P^\pi$ and initial states distribution $p_0$. Let $\mathcal{R}$ be the set of all recurrent states of the Markov chain and $\mathcal{R}_1, \ldots, \mathcal{R}_L$ the sets of recurrent classes, each associated with stationary distribution $\mu_l$. For each $s \in \mathcal{R}$, let $l(s)$ denote the index of the recurrent class to which $s$ belongs. Then, for any policy $\pi \in \Pi_S$, it holds that

$$f_K(\pi) - f_\infty(\pi) \geq \frac{c}{2K} \sum_{\substack{s \in \mathcal{R} \\ a \in \mathcal{A}}} \pi(a|s)^2 \left(\mu_{l(s)}(s)\right)^2 \operatorname*{Var}_{B \sim \mathrm{Ber}\left(\alpha_{l(s)}\right)} [B],$$

where $B \sim \mathrm{Ber}(p)$ denotes that $B$ is distributed according to a Bernoulli distribution such that $\mathbb{P}(B = 1) = p$ and

$$\alpha_{l(s)} = \lim_{t \to \infty} \mathbb{P}(S_t \in \mathcal{R}_{l(s)} | S_0 \sim p_0)$$

is the probability of absorption to recurrent class $\mathcal{R}_{l(s)}$ when $S_0 \sim p_0$.

*Proof.* For any policy $\pi \in \Pi_S$ it holds that

$$
\begin{aligned}
f_K(\pi) - f_\infty(\pi) &= \mathbb{E}_{\mathcal{T}_K}\left[f(\hat{d}_{\mathcal{T}_K})\right] - f\left(\mathbb{E}_{\mathcal{T}_K}\left[\hat{d}_{\mathcal{T}_K}\right]\right) \\
&\overset{(a)}{\geq} \frac{c}{2}\mathbb{E}_{\mathcal{T}_K}\left[\left\|\hat{d}_{\mathcal{T}_K} - d_\pi\right\|_2^2\right] \\
&= \frac{c}{2}\mathbb{E}_{\mathcal{T}_K}\left[\sum_{s\in\mathcal{S}, a\in\mathcal{A}}\left(\hat{d}_{\mathcal{T}_K}(s,a) - d_\pi(s,a)\right)^2\right] \\
&= \frac{c}{2}\sum_{s\in\mathcal{S},a\in\mathcal{A}}\mathbb{E}_{\mathcal{T}_K}\left[\left(\hat{d}_{\mathcal{T}_K}(s,a) - d_\pi(s,a)\right)^2\right] \\
&= \frac{c}{2}\sum_{s\in\mathcal{S},a\in\mathcal{A}}\mathrm{Var}_{\mathcal{T}_K}\left[\hat{d}_{\mathcal{T}_K}(s,a)\right] \\
&= \frac{c}{2}\sum_{s\in\mathcal{S},a\in\mathcal{A}}\mathrm{Var}_{\{T_1,\ldots,T_K\}}\left[\frac{1}{K}\sum_{k=1}^{K}\hat{d}_{T_k}(s,a)\right] \\
&= \frac{c}{2K}\sum_{s\in\mathcal{S},a\in\mathcal{A}}\mathrm{Var}_{T\sim\zeta_\pi}\left[\hat{d}_T(s,a)\right] \\
&= \frac{c}{2K}\sum_{s\in\mathcal{S},a\in\mathcal{A}}\left(\mathbb{E}_{T\sim\zeta_\pi}\left[\hat{d}_T(s,a)^2\right] - \mathbb{E}_{T\sim\zeta_\pi}\left[\hat{d}_T(s,a)\right]^2\right),
\end{aligned}
$$

where (a) follows from the strongly convex assumption and the fact that

$$
f(X) \geq f(\mathbb{E}[X]) + \nabla f(\mathbb{E}[X])^\top (X - \mathbb{E}[X]) + \frac{c}{2}\|X - \mathbb{E}[X]\|_2^2
$$
$$
\implies \mathbb{E}[f(X)] \geq f(\mathbb{E}[X]) + \frac{c}{2}\mathbb{E}\left[\|X - \mathbb{E}[X]\|_2^2\right],
$$

where $X$ is a random vector.

Focusing on the term $\mathbb{E}_{T\sim\zeta_\pi}\left[\left(\hat{d}_T(s,a)\right)^2\right]$ we have that

$$
\begin{aligned}
\mathbb{E}_{T\sim\zeta_\pi}\left[\left(\hat{d}_T(s,a)\right)^2\right] &= \mathbb{E}_{T\sim\zeta_\pi}\left[\left(\lim_{H\to\infty}\frac{1}{H}\sum_{t=0}^{H-1}\mathbf{1}\left(S_t = s, A_t = a\right)\right)^2\right] \\
&\overset{(a)}{=} \mathbb{E}_{Y_s}\left[(Y_s\pi(a|s))^2\right] \\
&= \pi(a|s)^2\mathbb{E}_{Y_s}\left[Y_s^2\right].
\end{aligned}
$$

Let $Z_H = \frac{1}{H}\sum_{t=0}^{H-1}\mathbf{1}\left(S_t = s, A_t = a\right)$. Step (a) above holds because: (i) from Lemma B.6 it holds that $\mathbb{P}\left(\lim_{H\to\infty} Z_H = Y_s\pi(a|s)\right) = 1$, i.e., $Z_H \to Y_s\pi(a|s)$ almost surely (note also that $|Z_H| \leq 1$ for all $H \in \mathbb{N}$); (ii) since $g(x) = x^2$ is continuous it holds that $g(Z_H) \to g(Y_s\pi(a|s))$ almost surely (note also that $|g(Z_H)| \leq 1$ for all $H \in \mathbb{N}$); (iii) from the bounded convergence theorem (Durrett, 2019, Theo. 1.6.7.) it holds that $\mathbb{E}_T\left[g\left(\lim_{H\to\infty} Z_H\right)\right] = \mathbb{E}_{Y_s}\left[g(Y_s\pi(a|s))\right]$. Also in (a), $Y_s$ is distributed according to the probability density function defined in Lemma B.5.

For the term $\mathbb{E}_{T\sim\zeta_\pi}\left[\left(\hat{d}_T(s,a)\right)\right]^2$, via similar arguments, it holds that

$$
\mathbb{E}_{T\sim\zeta_\pi}\left[\left(\hat{d}_T(s,a)\right)\right]^2 = \mathbb{E}_{Y_s}\left[(Y_s\pi(a|s))\right]^2 = \pi(a|s)^2\mathbb{E}_{Y_s}\left[(Y_s)\right]^2.
$$

Replacing both terms in our original lower bound yields

$$
\begin{aligned}
f_K(\pi) - f_\infty(\pi) &\geq \frac{c}{2K} \sum_{s \in \mathcal{S}, a \in \mathcal{A}} \pi(a|s)^2 \left( \mathbb{E}_{Y_s}\left[Y_s^2\right] - \mathbb{E}_{Y_s}\left[(Y_s)\right]^2 \right) \\
&= \frac{c}{2K} \sum_{s \in \mathcal{S}, a \in \mathcal{A}} \pi(a|s)^2 \mathrm{Var}\left[Y_s\right] \\
&\overset{(a)}{=} \frac{c}{2K} \sum_{s \in \mathcal{R}} \sum_{a \in \mathcal{A}} \pi(a|s)^2 \mathrm{Var}\left[Y_s\right] \\
&\overset{(b)}{=} \frac{c}{2K} \sum_{s \in \mathcal{R}} \sum_{a \in \mathcal{A}} \pi(a|s)^2 \mathrm{Var}_{B \sim \mathrm{Bernoulli}\left(p=\alpha_{l(s)}\right)} \left[\mu_{l(s)}(s)B\right] \\
&= \frac{c}{2K} \sum_{l=1}^{L} \mathrm{Var}_{B \sim \mathrm{Ber}(\alpha_l)} \left[B\right] \sum_{s \in \mathcal{R}_l} \sum_{a \in \mathcal{A}} \pi(a|s)^2 \mu_l(s)^2
\end{aligned}
$$

where in (a) we let $\mathcal{R}$ denote the set of all recurrent states for the Markov chain with transition matrix $P^\pi$ and the equality holds because if $s$ is transient then $\mathbb{P}(Y_s = 0) = 1$ and, hence, $\mathrm{Var}\left[Y_s\right] = 0$. In (b), the equality holds since we can rewrite random variable $Y_s$ using a scaled Bernoulli random variable. $\square$

## C. Empirical Results

In Algorithm 1 we present the pseudocode of the sampling scheme used to approximate the different GUMDPs formulations, under both discounted and average occupancies. The different objectives can be approximated, for a sufficiently large number of iterations $N$, by running Algorithm 1 with the desired $K$, $H$, and $\gamma$ parameters. Under the discounted setting, we vary parameters $K$, $H$, and $\gamma$. Under the average setting, we vary $K$ while setting $\gamma \approx 1$ and $H = \infty$. Under both the average and discounted setting, we set $K = \infty$ and $H = \infty$ to compute the infinite trials objective.

---

**Algorithm 1** Estimating $f_{K,H}(\pi)$ via samples.

---

1: **Inputs:** $N \in \mathbb{N}$ (num. of iterations), $K \in \mathbb{N}$ (num. of trajectories), $H \in \mathbb{N}$ (trajectories' horizon), and $\gamma$ (discount factor).
2: $\hat{f}_0 = 0$
3: **for** $n$ in $\{1, \dots, N\}$ **do**
4: $\quad \{\tau_1, \dots, \tau_K\} \sim \zeta_\pi$
5: $\quad \hat{d}_K(s,a) = \frac{1-\gamma}{1-\gamma^H} \frac{1}{K} \sum_{k=1}^{K} \sum_{t=0}^{H-1} \gamma^t \mathbf{1}(s_{k,t} = s, a_{k,t} = a), \ \forall s, a$
6: $\quad \hat{f}_n = \hat{f}_{n-1} + \frac{f(\hat{d}_K) - \hat{f}_{n-1}}{N}$
7: **end for**
8: **Return:** $\hat{f}_N$

---

### C.1. Empirical results for the GUMDPs in Fig. 1

We consider the GUMDPs depicted in Fig. 1, representative of three tasks in the convex RL literature. Under $\mathcal{M}_{f,1}$ we let $\pi(\text{left}|s_0) = \pi(\text{right}|s_0) = 0.5$, $\pi(\text{right}|s_1) = 1$ (zero otherwise), and $\pi(\text{left}|s_2) = 1$ (zero otherwise); for both $\mathcal{M}_{f,2}$ and $\mathcal{M}_{f,3}$ we let $\pi$ be the uniformly random policy. Figures 4, 5, and 6 display the results obtained under the different GUMDPs illustrated in Fig. 1 for different $\gamma$, $K$ and $H$ values. Figure 7 displays the results obtained for the three GUMDPs for different $K$ and $\gamma$ values with $H = \infty$.

### C.2. Empirical results for the GUMDPs in Fig. 1 with noisy transitions

We consider again the three GUMDPs illustrated in Fig. 1, but add a small amount of noise to the transition matrices of each GUMDP so that there is a non-zero probability, at each timestep, of transitioning from a given state to any other arbitrary state. Under $\mathcal{M}_{f,1}$ we let $\pi(\text{left}|s_0) = \pi(\text{right}|s_0) = 0.5$, $\pi(\text{right}|s_1) = 1$ (zero otherwise), and $\pi(\text{left}|s_2) = 1$ (zero otherwise); for both $\mathcal{M}_{f,2}$ and $\mathcal{M}_{f,3}$ we let $\pi$ be the uniformly random policy. Figures 8, 9, and 10 display the results obtained under the different GUMDPs illustrated in Fig. 1 for different $\gamma$, $K$ and $H$ values. Figure 11 displays the results obtained for the three GUMDPs for different $K$ and $\gamma$ values with $H = \infty$ and noisy transitions.

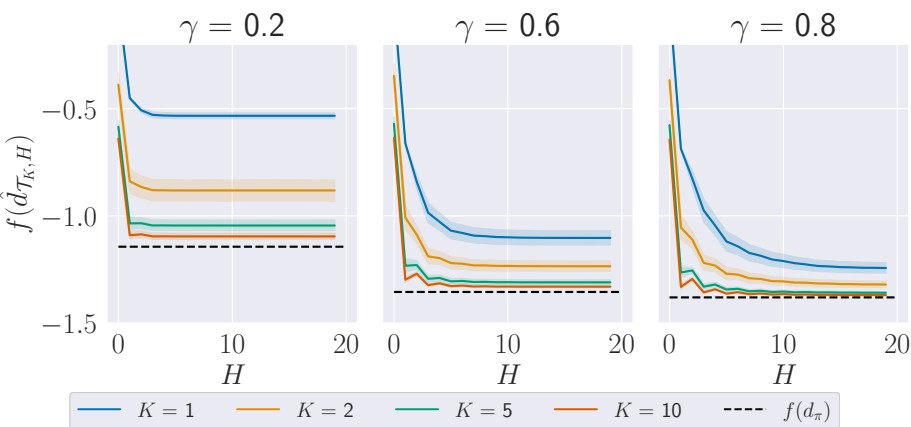

Figure 4: ($\mathcal{M}_{f,1}$, standard transitions) Empirical study of $f(\hat{d}_{\mathcal{T}_K,H})$ for different $K$, $H$ and $\gamma$ values under GUMDP $\mathcal{M}_{f,1}$ with policy $\pi(\text{left}|s_0) = 0.5$, $\pi(\text{right}|s_0) = 0.5$, $\pi(\text{right}|s_1) = 1$, $\pi(\text{left}|s_2) = 1$. The results are computed over 100 random seeds. Shaded areas correspond to the 95% bootstrapped confidence intervals.

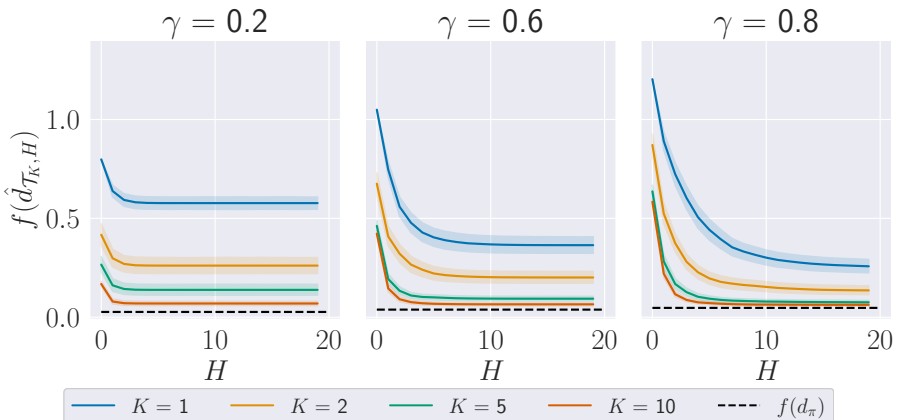

Figure 5: ($\mathcal{M}_{f,2}$, standard transitions) Empirical study of $f(\hat{d}_{\mathcal{T}_K,H})$ for different $K$, $H$ and $\gamma$ values under GUMDP $\mathcal{M}_{f,2}$ with a uniformly random policy. The results are computed over 100 random seeds. Shaded areas correspond to the 95% bootstrapped confidence intervals.

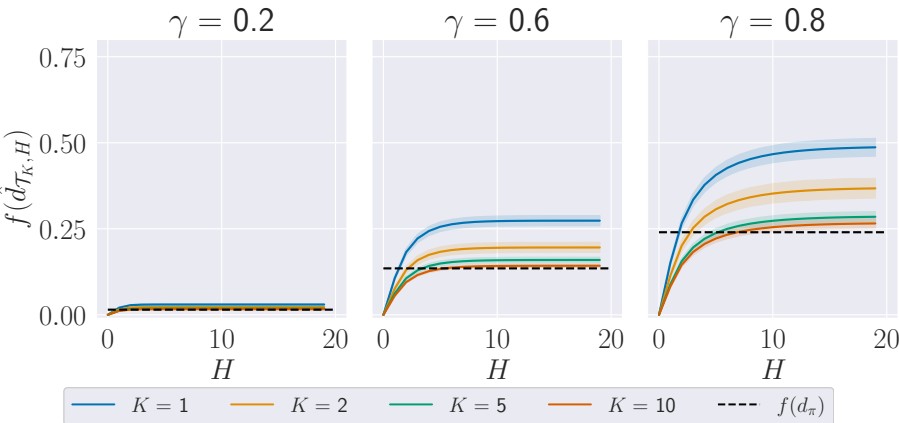

Figure 6: ($\mathcal{M}_{f,3}$, standard transitions) Empirical study of $f(\hat{d}_{\mathcal{T}_K,H})$ for different $K$, $H$ and $\gamma$ values under GUMDP $\mathcal{M}_{f,3}$ with a uniformly random policy. The results are computed over 100 random seeds. Shaded areas correspond to the 95% bootstrapped confidence intervals.

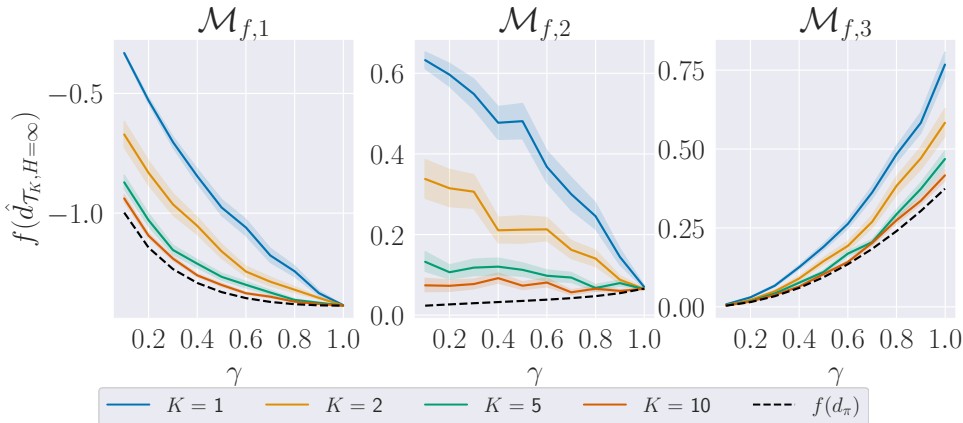

Figure 7: (Standard transitions) Empirical study of $f(\hat{d}_{\mathcal{T}_K, H=\infty})$ for different $K$ and $\gamma$ values with $H = \infty$. The results are computed over 100 random seeds. Shaded areas correspond to the 95% bootstrapped confidence intervals.

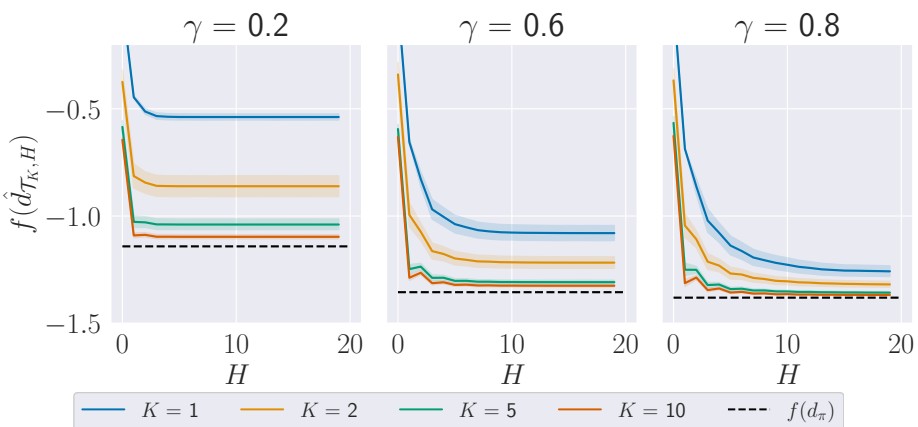

Figure 8: ($\mathcal{M}_{f,1}$, noisy transitions) Empirical study of $f(\hat{d}_{\mathcal{T}_K, H})$ for different $K$, $H$ and $\gamma$ values under GUMDP $\mathcal{M}_{f,1}$ with noisy transitions and policy $\pi(\text{left}|s_0) = 0.5$, $\pi(\text{right}|s_0) = 0.5$, $\pi(\text{right}|s_1) = 1$, $\pi(\text{left}|s_2) = 1$. The results are computed over 100 random seeds. Shaded areas correspond to the 95% bootstrapped confidence intervals.

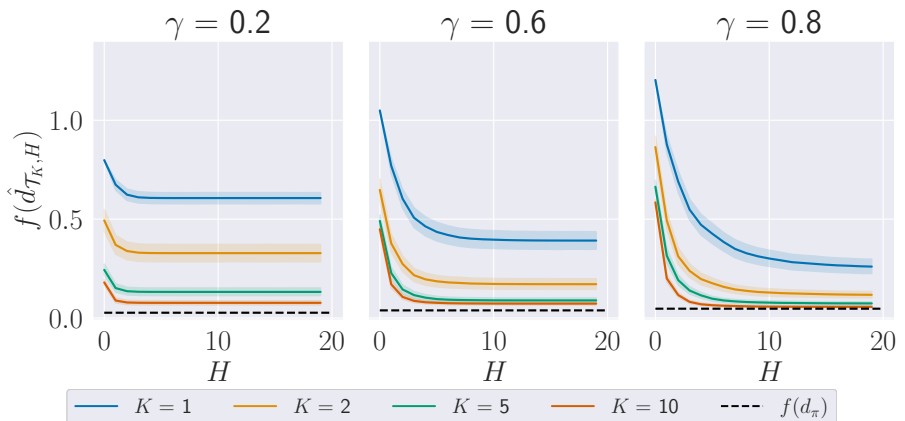

Figure 9: ($\mathcal{M}_{f,2}$, noisy transitions) Empirical study of $f(\hat{d}_{\mathcal{T}_K, H})$ for different $K$, $H$ and $\gamma$ values under GUMDP $\mathcal{M}_{f,2}$ with noisy transitions and a uniformly random policy. The results are computed over 100 random seeds. Shaded areas correspond to the 95% bootstrapped confidence intervals.

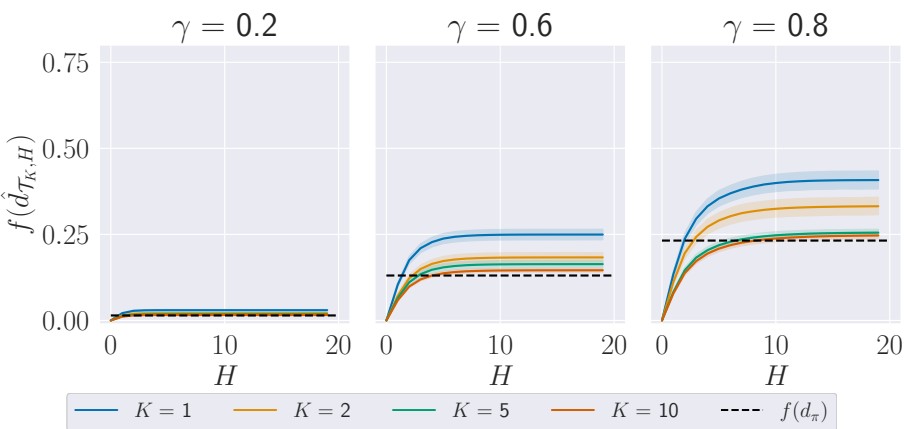

Figure 10: ($\mathcal{M}_{f,3}$, noisy transitions) Empirical study of $f(\hat{d}_{\mathcal{T}_K,H})$ for different $K$, $H$ and $\gamma$ values under GUMDP $\mathcal{M}_{f,3}$ with noisy transitions and a uniformly random policy. The results are computed over 100 random seeds. Shaded areas correspond to the 95% bootstrapped confidence intervals.

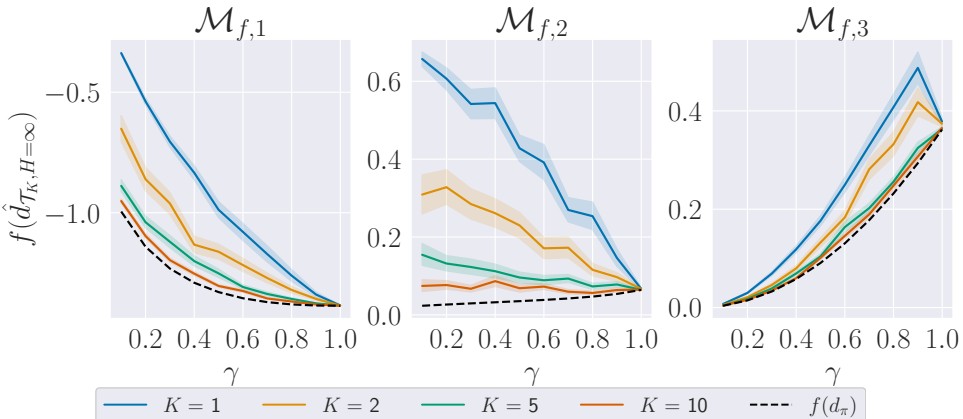

Figure 11: (Noisy transitions) Empirical study of $f(\hat{d}_{\mathcal{T}_K,H=\infty})$ for different $K$ and $\gamma$ values with $H = \infty$ and noisy transitions. The results are computed over 100 random seeds. Shaded areas correspond to the 95% bootstrapped confidence intervals.

