# OpenReview forum: "The Number of Trials Matters in Infinite-Horizon General-Utility Markov Decision Processes"
_ICML.cc/2025/Conference — ICML 2025 spotlightposter_

### Official Review · Reviewer_ntsy · 2025-03-13

**Overall Recommendation:** 4

**Summary:**

This paper investigates how the number of trials affects policy evaluation in infinite-horizon general-utility Markov Decision Processes (GUMDPs) for both the discounted and average cases. For the discounted case, the authors demonstrate that a mismatch generally exists between the finite- and infinite-trial formulations and provide lower and upper bounds to quantify this discrepancy. For the average case, they show that the structure of the underlying GUMDP also influences the mismatch. Finally, experiments on three simple GUMDPs validate the theoretical findings.

### update after rebuttal
The authors' rebuttal has addressed my questions. I keep my positive rating.

**Claims And Evidence:**

The claims made in the paper are well supported by clear proof and the experiment results.

**Essential References Not Discussed:**

None

**Experimental Designs Or Analyses:**

The experimental design follows previous work (e.g., the choice of GUMDPs), and the results align well with the theoretical findings.

**Methods And Evaluation Criteria:**

The proposed methods and benchmarks make sense for the problem.

**Other Comments Or Suggestions:**

* Line 377 (left): "Both $K$ and $H$ contribute to the tightness of the upper bound." However, under Theorem 4.3, the authors mention that "Finally, the upper bound does not get tighter as $H$ increases for fixed $K$." I may have missed something, but I would like to hear the authors' comments on this.
* Line 425 (left): It would be helpful if the authors could demonstrate how the equality holds for $\mathcal{M}_{f,2}$.

**Other Strengths And Weaknesses:**

**Strengths:**
- This paper is very well-written, easy to follow, and self-contained. Sections 2 and 3 provide extensive background to prepare readers for the later derivations, and Section 4 is clearly structured to present the main results.
- This work makes novel contributions to the study of GUMDPs, providing the first analysis of the impact of the number of trials.
- The experimental results align well with the theoretical findings.

**Questions For Authors:**

Please see comments to above questions

**Relation To Broader Scientific Literature:**

This work contributes to the study of GUMDPs, providing the first analysis of the impact of the number of trials.

**Theoretical Claims:**

I briefly checked the proof of Remark 3.1, 3.2, Theorem 4.1, 4.2 and 4.3. They look correct to me.

---

> ### Author Rebuttal · Authors · 2025-03-29
>
> We thank the reviewer for the positive review, comments, and concerns raised. We answer below the questions/concerns raised:
> - *"Line 377 (left): "Both $K$ and $H$ contribute to the tightness of the upper bound." However, under Theorem 4.3, the authors mention that "Finally, the upper bound does not get tighter as $H$ increases, for fixed $K$." I may have missed something, but I would like to hear the authors' comments on this."*: The aim of our work is to show that the number of trials, $K$, matters. In the context of discounted GUMDPs, to analyze the mismatch between the finite and infinite trials settings, we introduced a truncated estimator (equation 5) that depends on $K$, the number of trajectories, but also on $H$, the length of the sampled trajectories. With our comment after Theo. 4.3. ("the upper bound does not get tighter as $H$ increases, for fixed $K$"), we wanted to emphasize that the number of trials $K$ is the key parameter regulating the tightness of the bound, i.e., even if $H$ is "very high" so that the bias term $2\gamma^H$ is very small, if $K$ is low then we expect the upper bound to be loose; then, if we gradually increase $K$, the upper bound tightens with a rate of $1/\sqrt{K}$. I.e., setting $H$ to be high alone will not make the upper bound tight. However, naturally, for a fixed $K$, increasing $H$ makes the bias of the estimator to decrease and, hence, we write while commenting our experimental results that "both $K$ and $H$ contribute to the tightness of the upper bound.". However, this should not distract us from our key objective, which is to show that $K$, the number of trials, matters irrespective of the value of $H$. Our experimental results illustrate this (Fig. 2): for any fixed $K$, increasing $H$ decreases the bias of the estimator, but the gap between the infinite and finite trials formulations only disappears if $K$ is sufficiently high (as well as $H$ so that the bias is low). We commit to making this clearer in the final version of our manuscript.
> - *"Line 425 (left): It would be helpful if the authors could demonstrate how the equality holds for $\mathcal{M}_ {f,2}$."*: The intuition is that, for $\mathcal{M}_{f,2}$, the distribution of initial states is such that the agent always starts in $s_0$. Now, if the policy is stochastic in both states $s_0$ and $s_1$, it happens that the Markov chain induced by such a stochastic policy comprises a single recurrent class ($s_1$ is always reachable from $s_0$ and vice versa) and, hence, there exists no mismatch between the infinite and finite trials objectives (in light of our Theo. 4.4). Finally, for any deterministic policy, it happens that the agent always gets absorbed with probability one to the same recurrent class and, thus, the mismatch between the finite and infinite trials formulations fades away. More precisely, the only case where it is possible to have two recurrent classes is when the deterministic policy selects the right action in state $s_0$ and the left action in state $s_1$. However, since the agent always starts in $s_0$, recurrent class {$s_1$} is unreachable and, hence, there exists no mismatch between the finite and infinite trials formulations. This result agrees with our Theo. 4.6 since the probability of getting absorbed into one of the recurrent classes is one and the probability of getting absorbed to the other is zero and, hence, the lower bound is zero. We commit to making this clearer in the final version of our manuscript.
>
> We hope our answers addressed the reviewer's main concerns.

---

### Official Review · Reviewer_u79Y · 2025-03-14

**Overall Recommendation:** 4

**Summary:**

The paper analyzes the impact of the number of trails in estimating the objectives for GUMDPs. For both the discounted and average settings, it is shown by examples that there are mismatches between the finite-trial estimates and the actual infinite-trail objectives. Bounds on the mismatches are provided, with numerical results supporting the theoretical claims.

**Claims And Evidence:**

- Theorem 4.1 shows by an example that there is mismatch for the discounted setting in general.
- Theorem 4.2 and 4.2 provide lower and upper bounds on the mismatch for the discounted setting.
- Theorem 4.4 claims that there is no mismatch in the average setting if the problem is unichain under any policy.
- Theorem 4.5 shows by an example that there is mismatch for the average setting in general.
- Theorem 4.6 provides a lower bound on the mismatch for the average setting given the set of recurrent classes.
- The existence of mismatches and their trends are illustrated in the numerical results.

**Essential References Not Discussed:**

None

**Experimental Designs Or Analyses:**

Three simple GUMDPs are considered in the numerical experiments. Although they are simple problems, the results illustrate the mismatches in objective estimations. The noiseless vs noisy results provide an interesting observation for convergence and non-convergence of the mismatches when the problem is unichain or not.

**Methods And Evaluation Criteria:**

Proofs are provided for the theorems.

**Other Comments Or Suggestions:**

None

**Other Strengths And Weaknesses:**

None

**Questions For Authors:**

In Fig 3b, it looks like there are some discontinuities in the performance of $M_{f, 3}$ around $\gamma=0.9$ where the finite-trail performance seems to diverge away from the infinite-trail one, but then converges back to it. Is that expected from theoretical analysis?

**Relation To Broader Scientific Literature:**

From the results of this work, we should be cautious when doing finite-sample policy evaluation.

**Theoretical Claims:**

The proofs and proof sketches in the main text seem correct, though not checking all details in the supplementary.

---

> ### Author Rebuttal · Authors · 2025-03-29
>
> We thank the reviewer for the positive review, comments, and concerns raised. We answer below the questions/concerns raised:
> - *"In Fig 3b, it looks like there are some discontinuities in the performance of around where the finite-trail performance seems to diverge away from the infinite-trail one, but then converges back to it. Is that expected from theoretical analysis?*": In Fig 3 (b), in the right-most plot, we agree that there appear to exist some kind of discontinuities since, as $\gamma$ increases, some curves appear to start deviating more and more from the dashed line (infinite trials curve), eventually converging back together as $\gamma \approx 1$. We carried out a detailed analysis as to why this is the case and realized that this happens due to a particular interaction between the occupancy induced by the policy as we increase $\gamma$ and the objective function. Whether these discontinuities may appear depends on the particular GUMDP instance at hand and the policy considered. For example, as seen in Fig. 3 (b) for the other two GUMDPs, the trend is different as the mismatch fades away as $\gamma$ increases. Nevertheless, all the results in Fig. 3 (b) agree with our theoretical results: (i) for the discounted case $\gamma <1$, there exists, in general, a mismatch between the finite and infinite trials formulations, which fades away as the number of trajectories $K$ increases; and (ii) for the average setting ($\gamma \approx 1$), the finite and infinite trials formulations are equivalent since the noisy transition matrices enforce that the underlying GUMDPs have a single recurrent class (hence, being unichain) under the considered policies. Unifying the analysis of the discounted and average settings is beyond the scope of our work, and we envision several technical challenges will arise; even for the case of linear objectives/MDPs, such analysis can be challenging (e.g., the theory of Blackwell optimality).
>
> We hope our answers addressed the reviewer's main concerns.

---

### Official Review · Reviewer_dxJG · 2025-03-21

**Overall Recommendation:** 4

**Summary:**

This paper analyzes the impact of the number of sampled trajectories on the estimation of the return for infinite-horizon MDPs with a general utility function. The classical MDP setup corresponds to linear utility, and in this case there is no bias induced by considering only a finite number of sampled trajectory - the main result of this paper is to show that this does not hold for the case of general utility. The paper highlights this situation for the case of policy evaluation, and provides numerical experiments.

## update after rebuttal

I thank the authors for taking the time to respond to my comments. My overall assessment of the paper is positive and I will keep my score.

**Claims And Evidence:**

The proofs of the claims of this paper are sketched in the main body of the paper. More detailed proofs are provided in appendices.

**Essential References Not Discussed:**

N/A

**Experimental Designs Or Analyses:**

As detailed above, I am fine with the numerical experiments. Perhaps it would have been nice to have studied a more structured/realistic MDP instance but the paper already does a great job on the theory side.

**Methods And Evaluation Criteria:**

The numerical experiments are restricted to a very simple setup (an MDP instance with a handful of states, as illustrated in Figure 1) but it suffices to provide a clear picture of the effects studied in this paper.

**Other Comments Or Suggestions:**

* Why is there a minus sign in the definition of the expected discounted cumulative reward $\langle d_{\gamma,\pi}, -r\rangle$? Why not defining the optimization program on page 2 (arg min …) as a maximization? Is it because you’ll use convex functions $f$ afterward? Also, it should be $\pi* \in \arg \min$ instead of $\pi^* = \arg \min$ since $\arg \min$ is the set of all possible optimal policies.

* Redundant definition of item (i) before and after equation (3)

* In equation (3), is the minimum always attained, even among Markovian policies $\Pi_{M}$? Shouldn’t it be an $\inf$? Also, can you provide refernece here again for the optimality of stationary policies for the case of discounted GUMDPs + average unichain GUMDPs?

* Please recall the definition of $\zeta_\pi$ (line 194 + 200 + 210).

* Line 212: is the convergence $f_{K,H} = f_{K}$ as $H\rightarrow + \infty$ uniform over $\pi$?

**Other Strengths And Weaknesses:**

* Strengths: well-written, mathematical sound, good literature review

* Witness: the scope of the topic covered in the paper is quite restricted - only policy evaluation, and only the impact on the number of trials. Some findings appear trivial when stated in English, e.g. line 78: ``*We show, both theoretically and empirically, that the agent’s performance may depend on the number of infinite-length trajectories drawn to evaluate its performance*”.

**Questions For Authors:**

N/A --- see my comments above.

**Relation To Broader Scientific Literature:**

This paper is related to the growing literature on general utily MDPs. The paper does a good job at providing a concise literature review. To some extent the paper is based on demonstrating that a suggestion from another paper (Mutti et al. 2023) is wrong (line 61:  *Finally, the authors suggest that the difference between finite and infinite trials fades away under the infinite-horizon setting*.)

**Theoretical Claims:**

I checked the sketch of the proofs in the main body and they appear correct.

---

> ### Author Rebuttal · Authors · 2025-03-29
>
> We thank the reviewer for the positive review, comments, and concerns raised. We answer below the questions/concerns raised:
> - *"Minus sign in the definition of the expected discounted cumulative reward*": We focused on the case of minimization (hence, convex $f$), similarly to what has been done in previous works (e.g., [1]). It is equally possible to consider the problem of maximization (hence, concave $f$), as, naturally, both formulations are equivalent.
> - *"It should be* $\pi^* \in \arg \min $ *instead of* $\pi^* = \arg \min $ *since is the set of all possible optimal policies."*: We agree with the reviewer and will fix this in the new version of the manuscript.
> - *"Redundant definition of item (i) before and after equation (3)"*: We thank the reviewer for the suggestion. We will incorporate it into the final version of our manuscript.
> - *"In equation (3), is the minimum always attained, even among Markovian policies?"*: Under the average multichain setting, the set of possible occupancies attained by all Markovian (possibly non-stationary) policies corresponds to the closed convex hull of a given set of points (Theo. 8.9.3. in [2]). Thus, the minimum is always attained.
> - *"Also, can you provide a reference here again for the optimality of stationary policies for the case of discounted GUMDPs + average unichain GUMDPs?"*: We thank the reviewer for the suggestion. We will add a reference to the results in [2] that are of relevance.
> - *"Recall the definition of $\zeta_\pi$"*: We thank the reviewer for the suggestion. We will incorporate it into the final version of our manuscript.
> - *"Line 212: is the convergence $f_{K,H}  = f_K$ as $H \rightarrow \infty$ uniform over $\pi$"*: Yes.
>
> We hope our answers addressed the reviewer's main concerns.
>
> [1] - Zahavy, T., O’Donoghue, B., Desjardins, G., and Singh, S. -  Reward is enough for convex mdps. CoRR (abs/2106.00661), 2021.
>
> [2] - Puterman, M. L. Markov decision processes: discrete stochastic dynamic programming. John Wiley \& Sons, 2014.

---

> > ### Comment · Reviewer_dxJG · 2025-04-02
> >
> > I thank the authors for taking the time to respond to my comments. My overall assessment of the paper is positive and I will keep my score.

---

### Official Review · Reviewer_3gey · 2025-03-23

**Overall Recommendation:** 3

**Summary:**

The paper continues work in the area of the so-called general utility MDPs. The main contribution is the clarification that infinite horizon criteria will not close the "finite vs infinite trial gap" contrary to a suggestion by Mutti et al. (2023) who studied the gap in the finite horizon setting.

Specifically, the setting is as follows: Consider a decision maker who wants to pick a policy such that in $K$ independent trials, the expected utility assigned to the average of the empirical occupancy measures underlying $K$ infinitely long trajectories is maximized. Formally, if $d_k$ is the empirical occupancy measure (discounted or average) underlying trajectory $1\le k \le K$,

$\bar{d}_K = (d_1+\dots + d_K)/K$

and the goal is to maximize $\mathbb{E}_\pi[ f(\overline{d}_K) ]$ by choosing an appropriate policy $\pi$. Here, $f$ maps distributions over state-action pairs to reals. One motivation to study this problem is to model risk-sensitive decision makers whose objective can be captured by a utility assigned to a state-action distribution.

As $K\to \infty$, since under $\mathbb{P}_\pi$, $d_1,\dots,d_K$ are independent, by the law of large numbers,

$\bar{d}_K $ converges to

$\mathbb{E}_\pi[ d_1 ]$

with probability one as $K\to \infty$. Hence, by Taylor series expansion, when $f$ is smooth, $f(\overline{d}_K)$ converges to

$f(\mathbb{E}_\pi[ d_1 ])$

with probability one as $K\to\infty$. The gap that the paper refers to is the difference between $\mathbb{E}_\pi[ f(\overline{d}_K) ]$ and this latter expression.

The first result concern the discounted setting. Here, the first main result (Theorem 4.2) shows that the gap is lower bounded by an $\Omega(1/K)$ term for $f$ strongly convex. The second main result (Theorem 4.3) shows that even if we truncate trajectories after $H$ steps and $f$ is convex Lipschitz, the gap is upper bounded by $O(\sqrt{1/K}+\gamma^H)$. The first result only applies to stationary policies.

In the average cost setting, focusing on stationary policies, the first result (Theorem 4.4) shows that the gap is zero for unichain MDPs provided $f$ is continuous and bounded. Next, the authors point out this does not necessarily hold for multichain MDPs (Theorem 4.5) and finish up with a general lower bound on the gap that is similar to Theorem 4.2 that applies to multichain MDPs (Theorem 4.6).

Finally, empirical results illustrate the theoretical findings.

## update after rebuttal
My assessment did not change after the rebuttal. The paper answers a well-defined question, asked in a previous paper, and is well-written. It is of interest for the community in the sense that the precursor paper hinted at some answer and this paper clarifies in a rigorous fashion whether that hint was correct. Yet, this question raised in the previous paper does not feel like a well thought out question -- as I explained it beforehand. Because of the merits of the paper, I still recommend weak accept -- take the paper if there is enough room.

**Claims And Evidence:**

The general claim is that just because the horizon is infinite, the gap will not disappear. Theorems 4.2 and 4.6 make this precise. In addition, the empirical study also suggest that the gap exist. I think overall the evidence is strong.

**Essential References Not Discussed:**

None.

**Experimental Designs Or Analyses:**

No issues; it is OK to run on these small MDPs; the plots are reasonably chosen.

**Methods And Evaluation Criteria:**

The empirical study is reasonable and the results do make sense in the context of the paper.
One small methodological remark is that the authors somehow say they ran experiments with $H=infty$. How can this be? I suppose this was just approximated by taking $H$ large. Is this convincing? Well, Fig 2. suggests that this is a reasonable approach, but ultimately, I think it is better to admit that there is no way to study this empirically (or am I missing something)?

**Other Comments Or Suggestions:**

N.A.

**Other Strengths And Weaknesses:**

**Originality**: Fair. The results have not been known, published. I guess the lower bounds are neat.

**Significance**. Questions I would ask:
Q1) Should we care about the finite trial objective?
Q2) Should we care about the gap described?
Q3) Are we surprised that there is this gap?

On Q1: While the finite trial objective with finite-horizon problems moves things closer to realism, the finite trial objective with infinite-horizon problems lacks realism: Making infinitely maps in the MDP is just not something that is going to happen in our world with finite resources and fixed speed execution. So why study this problem? Of course, the follow-up to this is to ask why study the infinite horizon setting to start with and the answer there is that the infinite horizon setting is chosen for elegance, generality, insight, and is an idealization and as such it has its own role. Yet, idealization moves against realism, so we get a contradiction.

Another answer to Q1 (and answer to Q2, as well) is that the finite trial objective appears to lead to "hard" problems. It is OK to point this out, but is this very exciting? Will we get algorithms that perform significantly better under the finite trial objective than if we instead consider the infinite trial objective? I have some doubts; the results show that under reasonable conditions the algorithms in the latter category will pay a price of size $1/\sqrt{K}$. The constant may be large here, but it is hard to imagine practical scenarios where $K$ is small and the decision maker knows enough to run their finite trial optimizer to do significantly better than what is predicted by this gap.

On Q3: The gap is not too surprising. It is not hard to realize that infinite length trajectories will often "do not wash out all randomness". I am guessing Mutti et al. thought of the one nice case when this happens, hence their comment. Is it worth clarifying this? In as much as we care about the finite trial objective, yes.

**Novelty**: The results are definitely novel; the tools used in the proofs are what one expects to see. Again, the lower bounds are a bit more interesting.

**Clarity**: The paper is superbly well written, except for skipping over the questions raised above on why should we even care given that the infinite horizon setting is not even realistic and how can you even run experiments in this setting (this second is a very minor issue).

**Questions For Authors:**

Is there a way to justify looking at this problem given that the infinite horizon setting is an idealization and the whole motivation is to move closer to realism?

**Relation To Broader Scientific Literature:**

There is a sequence of papers studying these problems and the literature is very well cited as far as I know.

**Theoretical Claims:**

The claims are precise, they are believable. Theorem 4.3 (upper bound for discounted setting) is quite predictable; the strength is in the lower bounds. None of the results are surprising, but the lower bounds required some careful calculation. Skimming through the proofs of these, I believe they are correct.

---

> ### Author Rebuttal · Authors · 2025-03-29
>
> We thank the reviewer for the comments and concerns raised. We answer below the questions/concerns raised:
> - *"Is there a way to justify looking at this problem given that the infinite horizon setting is an idealization and the whole motivation is to move closer to realism?"*: We agree and acknowledge that the reviewer raises a relevant point, which leads to an interesting discussion. Still, we argue that there are key differences between the finite and infinite-horizon formulations and, therefore, we should see them as being complementary to each other.
>
>     First, the infinite-horizon setting has been thoroughly considered by previous works in the literature of GUMDPs/convex RL (e.g., [1] considers both discounted and average settings). Therefore, it is of great interest to make the research community aware that the infinite-horizon GUMDPs framework hides some implicit assumptions (infinite trials assumptions), and that the performance of a trained agent may significantly deviate (as we study in this work) at test time in comparison to the expected one. We also highlight that such a mismatch between the finite and infinite trials has been overlooked by previous research, as we describe in our Introduction.
>
>     Second, we argue the infinite-horizon framework is inherently different from the finite-horizon one and, thus, both should be studied in the context of GUMDPs (and, particularly, in the finite-trials regime). This is because infinite-horizon formulations allow to induce certain orders of preference over policies that are not so easily induced under a finite-horizon formulation. For example, one may want to use discounting to tradeoff between earlier/later costs/rewards (or in the case of GUMDPs, between the "importance" of states that are visited earlier or later in the trajectories). Also, infinite-horizon discounted GUMDPs can be used to model finite-horizon tasks where the length of the episode is uncertain (Sec. 5.3 in [2]). Therefore, we strongly believe that previous research adopted infinite-horizon formulations not only for their potential to simplify analysis (e.g., in the discounted setting, stationary policies are optimal) but also because they fundamentally differ from finite-horizon formulations in problem modeling, as exemplified above. Moreover, infinite-horizon formulations do not always simplify the problem yet remain widely used—e.g., in MDPs, the analysis of the average setting is arguably more intricate than the finite-horizon case. In conclusion, we view finite- and infinite-horizon formulations as complementary, both warranting study.
>
> - *"Should we care about infinite-finite trials gap described?"*     We argue that yes, we should care. For example, let us consider the single-trial case ($K=1$), which is very relevant in real domains where the agent may only be able to interact with the environment once (e.g., a robot/agent only has one life). In this case, the gap between the finite and infinite trials formulations can be significant, as our results showcase (constant $1/\sqrt{K}$ will not play a role here as it equals one). Even though our focus in this paper was on the task of policy evaluation, we agree with the reviewer that policy optimization in the single-trial setting is harder than in the case of infinite trials. However, we conjecture and have some preliminary evidence that it is, indeed, possible to do much better than the infinite trials policy in a reasonably efficient fashion for the single-trial setting. We leave such a study for future work.
>
> - *"On running experiments with $H = \infty$:"* As suggested by the reviewer, in practice, it is impossible to draw trajectories of infinite-length. With this in mind, when running our experiments, we tuned $H$ so as to mitigate the impact of trajectories' truncation in our experimental results. We did this by running our experiments with "sufficiently high" $H$ values and plotting the values of the objectives as we increase $H$ and looking at how fast the curves were stabilizing (similar to the curves in Fig. 2). This allowed us to have a good confidence that the picked $H$ makes the impact of trajectories' truncation negligible.
>
> We hope our answers addressed the reviewer's main concerns.
>
> [1] - Zahavy, T., O’Donoghue, B., Desjardins, G., and Singh, S. -  Reward is enough for convex mdps. CoRR (abs/2106.00661), 2021.
>
> [2] - Puterman, M. L. Markov decision processes: discrete stochastic dynamic programming. John Wiley \& Sons, 2014.

---

> > ### Comment · Reviewer_3gey · 2025-04-02
> >
> > Thanks, I appreciate the answers.
> > Some thoughts on the arguments brought forward in the rebuttal. First on whether the infinite-horizon setting is interesting.
> > * "has been thoroughly considered by previous works in the literature": Just because others previously studied something does not necessarily make it interesting. That they studied means the problem was interesting to them, but perhaps not for any real reason.. Anyhow, I know this argument is brought up a lot to justify relevance, but generally I find this a weak argument. (I am sure the authors are also aware of this).
> > * "mismatch between the finite and infinite trials has been overlooked by previous research": this does not seem to answer to the question whether the infinite horizon setting is interesting.
> > * "infinite-horizon framework is inherently different from the finite-horizon one" [..] hence it should be studied. Different does not make it interesting.
> > * "infinite-horizon discounted GUMDPs can be used to model finite-horizon tasks where the length of the episode is uncertain" this is a valid reason! perhaps related to all this pondering about why study the infinite-horizon setting: as far as I see the previous work that pointed out the finite trial gap -- in the finite horizon setting -- did not quantify the size of the gap there. In fact, looking at the proof of Theorem 4.2, the proof does not seem to use anything about whether the horizon is finite or infinite. This makes the paper perhaps a little more interesting. (Related to this: Theorems 4.2 and 4.6 seems to be special cases of something more general. Perhaps this is also worth to point out.)
> > * "infinite-horizon formulations do not always simplify the problem yet remain widely used" this does not answer the question either.
> > Second, on whether characterizing the size of the gap is interesting.
> > As suggested in the rebuttal, consider K=1. The gap in this case is large. Do we expect any *learning* algorithm to do well on the single trial objective? I guess we all would say no. This problem is "too hard". So the conclusion is perhaps that the single trial (or finite trial) objective, while it may look natural, will not be tractable in general. Knowing that the gap is large rules out some type of algorithms but it does not rule out others (for the single trial objective). So maybe as a first step towards understanding what can and cannot be done, one could analyze the gap (as done in this paper).

---

### Official Review · Reviewer_EizD · 2025-03-23

**Overall Recommendation:** 3

**Summary:**

This paper focus on General-utility MDP (GUMDPs) generalizes the MDPs framework by considering convex objective functions $f$ of the occupancy frequency $d$ induced by a given policy.  The authors analyze the discrepancy between the finite-trials formulation $f_K$ and the infinite-trials formulation $f_\infty$ in the context of infinite horizon discounted GUMDPs and average GUMDPs extending prior research by [1] that focus on finite horizon.

References:

[1] Mutti, Mirco, et al. "Convex reinforcement learning in finite trials." Journal of Machine Learning Research 24.250 (2023): 1-42.

**Claims And Evidence:**

The claims made in the submission seems to supported clearly based on the proof sketch and their experiment.

(1) **Remark 3.2**, analogous to Theorem 1 of [1], establishes that for a convex function $f$, the finite $f\_K$ and infinite $f\_\infty$ trials formulations differs for both discounted infinite GUMDPs and average criteria GUMDPs. This result is derived using Jensen’s Inequality: $f\_\infty(\pi) = f( \mathbb{E}[ \hat{d}\_{\mathcal{T}\_K} ] ) \leq \mathbb{E}[ f(\hat{d}\_{\mathcal{T}\_K}) ] = f\_K(\pi)$. Furthermore, **Theorems 4.1** and **4.5** use GUMDPs example (1c) to demonstrate that for multichain GUMDPs with a strictly convex objective function $f$, the inequality $f_K > f_\infty$ holds strictly for infinite horizon discounted GUMDP and average GUMDP respectively.

(2) **Theorem 4.2** extends remark 3.2 assuming c-strongly convex function $f$ for infinite horizon discounted GUMDPs, to provide a lower bound on $f\_K - f\_\infty$ via Jensen's gap bound.

(3) **Theorem 4.3**, extending Theorem 2 of [1], considers infinite-horizon discounted GUMDPs with an L-Lipschitz function $f$. It establishes a high-probability upper bound on the difference between the practical finite $K$-trail $H$-horizon value function and the true infinite-horizon value function $| f\_\infty - f(d\_{\mathcal{T}\_K,H}) |$ with Boole's and Hoeffding's inequality.

(4) **Theorem 4.4** in contrast to Remark 3.2, proves that when the average GUMDP is unichain and the continuous objective function $f$ is bounded then $f\_K = f\_\infty$.  This result is validated using the Ergodic theorem and properties of expectation.

(5) **Theorem 4.6** extends Lemma 3 of [1] to average GUMDPs, providing a lower bound on $f_K - f_\infty$ for c-strongly convex function $f$. The bound is expressed as the sum of the variance of the Bernoulli distribution of the recurrent states' probability.

References:

[1] Mutti, Mirco, et al. "Convex reinforcement learning in finite trials." Journal of Machine Learning Research 24.250 (2023): 1-42.

**Essential References Not Discussed:**

N/A

**Experimental Designs Or Analyses:**

The authors' experiments primarily focus on simple tabular examples with varying values of $K$ trails and $\gamma$ discount, and these seem to be correct. However, the analysis is limited to small problems, and it would be beneficial to see how the value function mismatch in larger domains.

**Methods And Evaluation Criteria:**

The authors support their claims with three simple GUMDP examples. The example in Figure 1c is well-explained, and Theorems 4.1, 4.5, and Figure 3 effectively illustrate the discrepancy between the finite and infinite trial formulations with this example.

However, the purpose of Figures 1a and 1b is unclear. While the figure caption states that all GUMDPs have deterministic transitions, the states in Figures 1a and 1b appear to have two distinct transitions each, raising ambiguity about whether these transitions correspond to different actions. Additionally, the authors assert that all GUMDPs in Figure 1 are multichain, but this is not explicitly justified. Providing a clearer explanation of what Figures 1a and 1b represent would help readers better understand their purposes.

**Other Comments Or Suggestions:**

The paper would be in a stronger position if it explicitly specified which aspects are identical to [1] and which parts of the extension present significant challenges. As it stands, it closely resembles [1] and applies similar techniques to extend the analysis from finite-horizon to infinite-horizon discounted and average GUMDPs.

References:

[1] Mutti, Mirco, et al. "Convex reinforcement learning in finite trials." Journal of Machine Learning Research 24.250 (2023): 1-42.

**Other Strengths And Weaknesses:**

Other Weakness:

1. Note that this paper is an extension of the finite horizon work in [1] to analysis of infinite horizon discounted and average GUMDPs. The main weakness of the paper lies in its novelty, as it appears quite similar to [1] and uses many of the same techniques to derive its theorems. The paper does not clearly specify which aspects of the extension from [1] are particularly challenging.

Other Strengths:

1. The paper does offer a new insight in Theorem 4.4, showing that unichain GUMDPs satisfy $f\_K = f\_\infty$ in average GUMDP.

2. The paper is well-written, with good structure and readability.

References:

[1] Mutti, Mirco, et al. "Convex reinforcement learning in finite trials." Journal of Machine Learning Research 24.250 (2023): 1-42.

**Questions For Authors:**

Please address the questions :

(1) Which aspects are identical to [1] and which parts of the extension present significant challenges.

(2) The figure caption states that all GUMDPs have deterministic transitions, yet Figures 1a and 1b depict states with two distinct transitions each. It would be helpful to clarify what these transitions represent.

**Relation To Broader Scientific Literature:**

Although this work is similar to [1], which focuses on finite-horizon GUMDPs, it provides valuable insights into the convergence properties of infinite-horizon discounted GUMDPs and average GUMDPs.

References:

[1] Mutti, Mirco, et al. "Convex reinforcement learning in finite trials." Journal of Machine Learning Research 24.250 (2023): 1-42.

**Theoretical Claims:**

It is unclear how the Hoeffding style inequality in the proof sketch and the full proof of Theorem 4.3 follows from Lemma 16 in [1], that proves the feasibility of the optimal policy. A more explicit explanation or a clearer connection between these results would help clarify the derivation.

References:

[1] Efroni, Y., Mannor, S., and Pirotta, M. Exploration-exploitation in constrained mdps. CoRR, abs/2003.02189, 2020.

---

> ### Author Rebuttal · Authors · 2025-03-29
>
> We thank the reviewer for the comments and concerns raised. We answer below each of the questions/concerns raised:
> - *"(1) Which aspects are identical to [1] and which parts of the extension present significant challenges.":* We thank the reviewer for asking about the differences between our article and [1], particularly regarding the technicalities of our results and proofs. Our analysis fundamentally differs, from a technical point of view, from that in [1] where the authors consider the finite-horizon case. The key difference is due to the fact that discounted and average occupancies are inherently different than occupancies induced under the finite-horizon setting. The fact that the occupancies are defined over trajectories of infinite length forced us to seek alternative ways to prove our results. This is because we are mostly worried about the long-term behavior of the Markov chain induced by the policy, and not focused on a fixed number of interaction steps as considered in [1].
>
>     The kind of results we provide also significantly differ from those in [1]: we are the first work to provide lower bounds on the mismatch between the finite and infinite trials objectives, as [1] only proves upper bounds in the context of policy optimization. Also, while [1] shows that, in general, there exists a mismatch between the finite and infinite trials formulations under finite-horizon settings, we prove a result that states that under certain conditions (the GUMDP being unichain), there exists no mismatch. Hence, the nature of our results and those in [1] greatly differ.
>
>
>     Finally, although Theo. 4.3. is indeed related to Theo. 2 in [1], we focused on the policy evaluation case and had to incorporate the fact that the trajectories are discounted. All our other theorems (Theo. 4.1, 4.2, 4.4, 4.5, and 4.6) are not connected at all to any result in [1], relying on significantly different proof techniques and lines of reasoning than those considered in [1]. For example, Theo. 4.2 and 4.6 rely on the (quite general) strongly convex assumption that was not considered in [1]; Theo. 4.4. relies on multiple Lemmas (Appendix C.2.) related to the study of the ergodicity of the Markov chain induced when conditioning the GUMDP with a given policy.
>
>     We commit to clarify in the final version of the article the novel aspects of our work in comparison to [1].
>
> - *"(2) The figure caption states that all GUMDPs have deterministic transitions, yet Figures 1a and 1b depict states with two distinct transitions each. It would be helpful to clarify what these transitions represent."*: We apologize for any confusion our visual depiction of the GUMDPs may have caused. We will improve this in the final version of our paper. Essentially, the GUMDPs we depict in Fig. 1 all have deterministic transitions. In Figs. 1 (a) and 1 (b), the different transitions coming out of the state nodes encode the transitions associated with each of the two actions (we commit to explicitly adding the action labels to the arrows to make it clearer). Finally, in the experimental section of our work (Sec. 4.3), we considered stochastic variants of the GUMDPs depicted in Fig.1. We consider the same GUMDPs as depicted in Fig. 1, but add a small amount of noise to the transition matrices so that there is a non-zero probability of transitioning to any other arbitrary state. Naturally, for the case of noisy transitions the visual depiction of the GUMDPs would be slightly different as the transitions are now stochastic, but we decided to present only the illustration of the GUMDPs for the case of deterministic transitions to make the illustrations easier to parse.
>
> We hope our answers addressed the reviewer's main concerns.

---

> > ### Comment · Reviewer_EizD · 2025-04-02
> >
> > I have reviewed the authors' response and appreciate their clarifications. Based on their rebuttal, we have adjusted our recommendation accordingly.
> >
> > Although the paper is not groundbreaking in terms of novelty, primarily extending finite-horizon GUMDP analysis to discounted infinite-horizon and average criteria. However, it is worth to note that they employ distinct proof techniques and steps to establish their bounds and conclusions due to the distinction of the occupancies for their settings. Additionally, the paper provides valuable insights, notably showing that the mismatch vanishes when the average GUMDP is unichain and the continuous objective function is bounded.

---

### Official Review · Reviewer_EP7E · 2025-03-25

**Overall Recommendation:** 3

**Summary:**

This paper analyzes infinite-horizon GUMDPs, highlighting the critical role of the number of sampled trajectories in policy evaluation. The authors theoretically and empirically demonstrate that, unlike standard MDPs, GUMDP performance metrics significantly depend on the number of trials, presenting theoretical lower and upper bounds for both discounted and average settings. They establish conditions under which finite and infinite trial objectives mismatch, particularly emphasizing multichain versus unichain structures.

**Claims And Evidence:**

Yes.

**Essential References Not Discussed:**

No.

**Experimental Designs Or Analyses:**

Experiments rely solely on illustrative toy scenarios with deterministic and artificially simplified state-action transitions. This makes the empirical validation questionable for practical GUMDPs in complex environments.

**Methods And Evaluation Criteria:**

Yes.

**Other Comments Or Suggestions:**

See other parts.

**Other Strengths And Weaknesses:**

See other parts.

**Questions For Authors:**

1. Have the authors considered how their theoretical results hold when relaxing assumptions on utility function convexity or continuity, which might often be violated in real-world scenarios?

2. Could the authors provide additional empirical analysis with more realistic, stochastic environments (beyond the simplistic toy examples), to demonstrate clearly the practical significance of their theoretical bounds?

**Relation To Broader Scientific Literature:**

The paper extends and significantly clarifies the conditions under which finite versus infinite trials matter in GUMDPs, directly complementing prior theoretical work. It effectively highlights differences in utility evaluations across trial conditions, an issue previously underexplored in the literature.

**Theoretical Claims:**

The paper assumes strongly convex and continuous utility functions with bounded domains. While mathematically convenient, realistic utility functions might violate these assumptions, potentially limiting theoretical findings’ practical relevance.

---

> ### Author Rebuttal · Authors · 2025-03-29
>
> We thank the reviewer for the comments and concerns raised. We answer below the questions/concerns raised:
>
> - *"Have the authors considered how their theoretical results hold when relaxing assumptions on utility function convexity or continuity, which might often be violated in real-world scenarios?''* We thank the reviewer for raising the discussion on how our assumptions may limit the generality of our results. Our assumptions remain valid under the great majority of objectives in the convex RL/GUMDPs literature and, hence, our results readily apply to many different tasks such as maximum state entropy exploration or imitation learning. We address each of our assumptions below:
>    1. **Strongly convex objectives:** We require the strong convexity assumption to prove the lower bounds of Theo. 4.2 and 4.6. All objective functions we consider in our work, which we believe to be representative of common objective functions used in the literature (e.g., maximum state entropy exploration, imitation learning, etc.), satisfy this assumption. We refer to Appendix C for the proofs of strong convexity. Naturally, if the objective is linear, as is the case for standard MDPs/RL, the objective function is not strongly convex; **however**, it should be clear that in such a case there does **not** exist a mismatch between the finite and infinite trials objectives and, thus, proving bounds on the mismatch between the objectives is meaningless. Finally, if our convexity assumptions are not met (e.g., the function is non-convex), we conjecture that the mismatch between the finite and infinite trials continues to exist in general, but we leave the study of such a case for future work.
>    2. **Continuous and bounded objectives:** We use the assumption that the objective is bounded and continuous to prove Theo. 4.4 and 4.6, which rely on the bounded convergence theorem. All objective functions we consider in our work are continuous and, up to the authors' knowledge, this assumption is satisfied for every objective function previously considered in the literature of GUMDPs. Thus, we believe our assumption is as general as it can be. Regarding the boundedness of the objective function, several objective functions (e.g., entropy and norm-based objectives) satisfy this assumption, as we state in our article. In the case of imitation learning tasks with KL-divergence-based objectives, as stated in the text before Theo. 4.4., we need to ensure that occupancy $d_\beta$ (the occupancy induced by the policy we want to imitate) is lower-bounded to meet our assumption. Nevertheless, we argue that: (i) this assumption is commonly used by previous theoretical works; and (ii) in practice, one can ensure the distribution is lower-bounded by adding a small value ($\epsilon$) to every entry of the occupancy. Finally, we highlight that boundedness assumptions are commonly considered in the literature of MDPs/RL.
>
> - *"Could the authors provide additional empirical analysis with more realistic, stochastic environments (beyond the simplistic toy examples), to demonstrate clearly the practical significance of their theoretical bounds?"*
>    1. **Simplistic environments/experiments:** While our GUMDPs comprise a small set of states, we argue they are representative of several tasks considered by previous works in the literature of convex RL/GUMDPs (we refer to Fig. 1 for a description of our GUMDPs). Also, we aimed to provide the simplest possible GUMDPs/experimental settings to stress the generality of our results, as acknowledged by reviewers 3gey and dxJG. We mainly used our empirical results to validate the theoretical results of our paper, which readily apply to GUMDPs of any dimension under the assumptions stated by each result, which we believe to be general enough to be representative of the great majority of objectives used by previous works.
>    2. **Experiments with stochastic transitions:** We also provide experiments with stochastic transitions in the paper (Fig. 3 (b)) and analyzed how the stochasticity of the transitions impacts our results, connecting our experimental results with the theoretical findings.
>
> We hope our answers addressed the reviewer's main concerns.

---

> > ### Comment · Reviewer_EP7E · 2025-04-06
> >
> > Thank you to the authors for their rebuttal. After considering the response, I have decided to maintain my original score.

---

### Decision · Program_Chairs · 2025-05-01

**Decision:**

Accept (spotlight poster)

**Comment:**

The reviewers uniformly agree that this is an original, well-written, and novel work. There are no concerns about the correctness of the work. The only concerns are relatively minor. I urge the authors to address the concerns about the works significance when preparing the final version of the work.